# Live slow-frozen human tumor tissues viable for 2D, 3D, ex vivo cultures and single-cell RNAseq

Gaetana Restivo[1,7], Aizhan Tastanova [1,7], Zsolt Balázs[2,7], Federica Panebianco[3], Maren Diepenbruck[3], Caner Ercan [4], Bodgan-T. Preca[3], Jürg Hafner[1], Walter P. Weber[5], Christian Kurzeder[5], Marcus Vetter[5], Simone Münst Soysal[5], Christian Beisel [6], Mohamed Bentires-Alj [3], Salvatore Piscuoglio[3,4], Michael Krauthammer[2] & Mitchell P. Levesque [1✉]

Biobanking of surplus human healthy and disease-derived tissues is essential for diagnostics and translational research. An enormous amount of formalin-fixed and paraffin-embedded (FFPE), Tissue-Tek OCT embedded or snap-frozen tissues are preserved in many biobanks worldwide and have been the basis of translational studies. However, their usage is limited to assays that do not require viable cells. The access to intact and viable human material is a prerequisite for translational validation of basic research, for novel therapeutic target discovery, and functional testing. Here we show that surplus tissues from multiple solid human cancers directly slow-frozen after resection can subsequently be used for different types of methods including the establishment of 2D, 3D, and ex vivo cultures as well as single-cell RNA sequencing with similar results when compared to freshly analyzed material.

[1] Department of Dermatology, University Hospital Zurich, University of Zurich, Zurich, Switzerland. [2] Department of Quantitative Biomedicine, University Hospital Zurich, Zurich, Switzerland. [3] Department of Biomedicine, Department of Surgery, University Hospital Basel, University of Basel, Basel, Switzerland. [4] Institute of Medical Genetics and Pathology, University Hospital Basel, Basel, Switzerland. [5] Department of Surgery, Breast Center, University Hospital Basel, Basel, Switzerland. [6] Department of Biosystems Science and Engineering, ETH Zurich, Basel, Switzerland. [7] These authors contributed equally: Gaetana Restivo, Aizhan Tastanova, Zsolt Balázs. ✉email: Mitchell.Levesque@usz.ch

Fresh surplus tissue material from surgery is often directly processed for downstream analysis, but this is not always possible due to time constraints, the absence of resources, or complicated logistics. Therefore, clinically relevant tissue samples that are preserved in biobanks are mostly stored as formalin-fixed paraffin-embedded (FFPE), optimum cutting temperature (OCT) blocks, or snap-frozen[1]. While tissues preserved in those conditions are an important resource for research, their use is limited to applications such as immunostaining or DNA/RNA isolation. In the last decades, several tissue cryopreservation methods have been developed for different research applications. In 2015, Alkema et al.[2] showed that ovarian patient-derived xenografts (PDXs) can be efficiently stored in fetal calf serum (FCS) and dimethyl sulfoxide (DMSO) medium, and this method preserves the tumor morphology and gene copy number after reinjection in mice. In 2016, Walsh et al.[3] showed that it is possible to generate organoids from slow-frozen BT474 xenograft tumors and that those have comparable viability and drug response of organoids generated from fresh tissues. Many studies with cryopreserved material from breast cancer have been published with promising applications for ex vivo drug testing[4–6]. Moreover, in 2019 Obara et al.[7] found that pluripotent stem cells derived from intact slow-frozen hair follicles can sustain multilineage differentiation potential. Single-cell RNA sequencing (scRNAseq) can also be applied on live slow-frozen material with similar results as when applied on fresh material. A recent study compared the variability in cell composition and cell-specific gene expression in the skin of patients with localized scleroderma utilizing scRNAseq. They tested cryopreserved samples (utilizing CryoStor® CS10) or fresh samples simply stored in RPMI. They observed similar yields and cell viability levels and a high correlation of gene expression between samples[8]. Wu et al.[9] applied scRNAseq to breast cancer, melanoma, and prostate cancer samples and showed that all cell clusters were recovered in cryopreserved samples and that biological processes were expressed to similar extents in fresh and cryopreserved samples. However, they did observe different cell cluster compositions in fresh and cryopreserved samples and noted that cryopreservation affected the various cell clusters differently, plasmablasts being affected the most. They also noted that cell suspensions were less affected by the freezing process. The greatest difference between slow-frozen and fresh samples was detected in the activation of stress pathways and the expression of heat-shock proteins[9]. Among various cryopreservation methods, DMSO-based slow-freezing has shown to be the most robust protocol to recover viable cells for scRNAseq with minimum background ambient RNA[10]. In this study, human biopsies from different solid cancers were slow-frozen directly after surgery using the classical cryopreservation method with FCS and 10% DMSO and applied a broad range of methods including 2D, 3D, and ex vivo cultures establishment and scRNAseq from different solid tumors. We report here a retrospective analysis and the success rate of establishing primary cell cultures obtained from fresh or slow-frozen melanoma biopsies. Moreover, we demonstrate the feasibility to obtain organoids from slow-frozen tissue comparable to freshly processed material from colorectal cancer (CRC). We also succeeded in cultivating ex vivo shave biopsies of basal cell carcinoma (BCC) after slow-freezing, showing a comparable amount of viable and proliferating BCC tumor cells when compared to shave biopsies cultivated directly after resection. We compared scRNAseq results from fresh and slow-frozen material derived from BCC, CRC biopsies, and fine-needle aspirates (FNA) of a melanoma metastasis (overview in Fig. 1). Using scRNAseq, we demonstrated a comparable recovery rate, cell type composition, as well as cell type proportion in paired fresh and slow-frozen clinical samples. Overall, we propose how transformative technology can change the standard practice of biobanking by directly comparing fresh and slow-frozen cancer types and different applications.

## Results

In our study, we directly compared slow-frozen versus fresh tissue from BCC, CRC, and melanoma (see Table 1 for a detailed description of samples and application). As depicted in the project workflow in Fig. 1, cancer tissue was either cut into small pieces (BCC, CRC) or collected as shave biopsies (ex vivo tumor) (BCC). The small tumor pieces (approx. 2 x 2 mm) were either directly digested to a single-cell suspension (BCC, CRC) (F) or slow-frozen (BCC, CRC) (S) and subsequently used for 2D and 3D culture establishment and scRNAseq. Shave biopsies were either used fresh (F) or slow-frozen (S) and used to establish ex vivo cultures. Moreover, melanoma FNAs were either directly processed for scRNAseq or slow-frozen and subsequently used for scRNAseq. The storage duration of slow-frozen samples paired to freshly processed samples in this study was between 1 week to 14 months for scRNAseq, <1 month to 7 months for samples processed for 2D (with no significant difference in time of storage between successful and not-successful cultures) and 1 week for 3D and ex vivo cultures (Supplementary Table 1).

### Retrospective analysis of primary cell culture establishment from fresh and slow-frozen melanoma tissue.

Slow-frozen tissue biopsies can be used for several applications including the isolation and establishment of cell cultures. In order to compare the success rate of 2D cell culture, we performed a retrospective analysis on cells isolated either from fresh or slow-frozen melanoma tissue (not paired). Melanoma cell cultures were considered successful when the mutations present in the parental tumors were present in the isolated cells and for at least two passages and if the contamination of non-melanoma cells (e.g., fibroblasts) was less than 10%. Non-melanoma cells contamination was reduced by differential trypsinization or by inducing senescence through contact inhibition, serum starvation, or deprivation of adhesion. Non-melanoma cells contamination was assessed by morphological examination and derived from the percentage of cells carrying the specific mutations (found in the parental tumors) in the cell culture populations (cells carrying the mutations are melanoma, cells negative for the mutations are non-melanoma cells). As shown in Fig. 2a, the success rate of cells isolated from fresh and slow-frozen melanoma tissues is comparable. We analyzed 35 cell cultures obtained from fresh material and 36 obtained from slow-frozen tissue (not paired). For both groups we obtained 22 successful cell cultures (success rate fresh = 63%, slow-frozen = 60%). In Fig. 2b, we show an example of a successful melanoma cell line (top picture) and an unsuccessful one (bottom picture) in which non-melanoma cells contamination was present.

### 3D culture of CRC from fresh and slow-frozen material.

Recently, 3D culture technology has led to the development of novel cancer models. Accumulating evidence indicates that patient-derived organoids (PDOs) represent important modeling tools in the pre-clinical investigation of malignancies in predicting drug response[11,12]. It has been shown that PDO-3D cultures preserve the histological, genetic, and phenotypic features of the parental tumor and maintain its heterogeneity, allowing their application in multiple research fields[13,14]. We compared the success rate of PDO generation and histopathological features from fresh and cryopreserved matched tissues derived from two different stage IV CRC patients. First, we compared cell viability in fresh and matched slow-frozen tissue samples and observed no substantial difference (cell viability rates 50% to 55%,

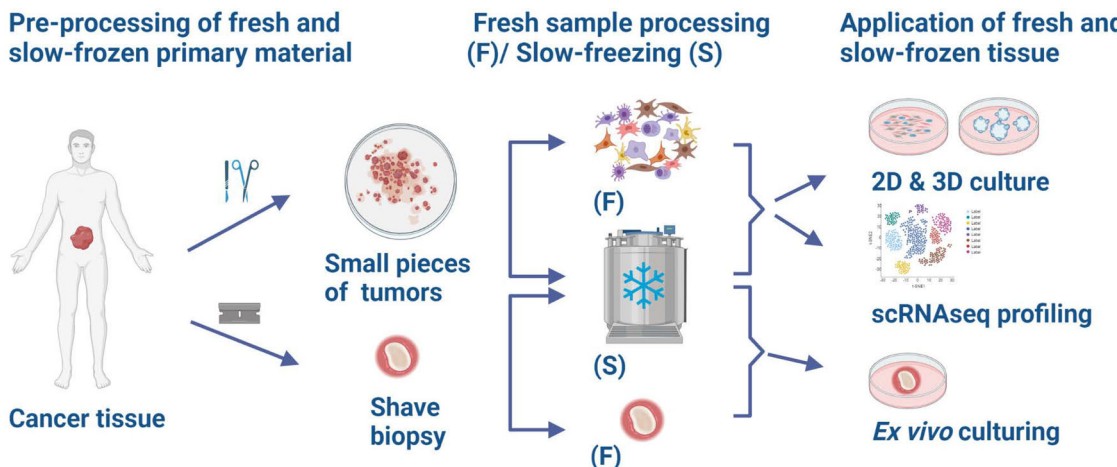

**Fig. 1 Schematic representation of the project workflow.** Cancer tissue was either cut into small tumors (BCC, CRC) or collected as shave biopsies (BCC). The small tumor pieces were either immideately digested to single-cell suspension (BCC, CRC) (F) or slow-frozen (BCC, CRC) (S) and subsequently used for 2D and 3D culture establishment and scRNAseq. Shave biopsies were either used fresh (F) or slow-frozen (S) and used to establish ex vivo cultures. Moreover, melanoma FNAs were either slow-frozen or directly processed for scRNAseq. (Created with BioRender.com).

| Cancer type | Sample ID | Sample Type | Diagnose | Age | Sex | Location | Application |
|---|---|---|---|---|---|---|---|
| Basal cell carcinoma | BCC_5613 | Resection | Nodular/sclerotic | 59 | F | Nose | scRNAseq |
| Basal cell carcinoma | BCC_5679 | Resection | Superficial/Micronodular | 84 | M | Nose | scRNAseq |
| Basal cell carcinoma | BCC_5680 | Resection | Superficial/Sclerotic/Nodular | 66 | M | Neck | scRNAseq |
| Basal cell carcinoma | SB 01 | Shave Biopsy | Not defined | 67 | M | Not defined | 3D ex vivo culture |
| Basal cell carcinoma | SB 02 | Shave Biopsy | Nodular/Sclerotic | 76 | F | Nose | 3D ex vivo culture |
| Colon primary | P83 | Resection | IIa (pT3, pN0, L0, V1, Pn1, R0, Bd3) | 74 | F | Colon | scRNAseq |
| Colon primary | P86 | Resection | IIa (pT3, pN0, L0, V0, Pn0, R0) | 54 | M | Colon | scRNAseq |
| Colon metastasis in liver | P117 | Resection | IVa (pT4a, pN2a, cM1, L1, V1, Pn0, R0, Bd3) | 51 | F | Liver | 3D Organoids |
| Colon primary | P134 | Resection | IVa (pT2 pN0, L0 V0 Pn0 R0, cM1a) | 72 | F | Colon | 3D Organoids |
| Melanoma | FNA_M_1 | Fine-needle aspirate | Metastatic melanoma BRAF p. Val600Lys | 73 | M | Trunk | scRNAseq |

**Table 1 Sample overview and different methods used.**

Supplementary Table 2). Those results are in line with other publications, where no difference in cell viability was observed in fresh versus slow-frozen tissue[15]. Then, we monitored organoid development and morphology for 4 weeks in 3D culture. We were able to generate organoids from fresh and matched slow-frozen tissues after 4 days and 8–10 days of culturing, respectively. After the organoids were generated, we did not observe differences in growth and morphology by bright-field microscopy. We assessed the expression of the homeobox protein CDX2, which is a transcription factor responsible for the differentiation and maintenance of the intestinal phenotype[16] and keratin 20 (CK20), whose expression is observed consistently higher in the well-differentiated CRC tissue samples[17] (Fig. 2c and Supplementary Fig. 1a).

**Ex vivo culture of BCC from fresh and slow-frozen material (shave biopsies).** Other models that can be used for testing new drugs are ex vivo tissue slices[18]. We compared the histopathological features of shave biopsies from fresh and slow-frozen matched tissues derived from two different BCC patients (Fig. 2d and Supplementary Fig. 1b). Histopathological examination using hematoxylin and eosin (H&E) is the gold standard to confirm the clinical diagnosis of BCC. However, histopathological examination by H&E alone does not accurately diagnose and distinguish some types of BCC which can be confused with other types of

carcinomas. To better characterize the tumor, we used antibodies against BerEP4 and the proliferation marker Ki67. BerEP4 is a monoclonal antibody that is currently used as a marker of BCC. It works by binding to the EpCAM antigen, which is a transmembrane epithelial glycoprotein cell adhesion molecule[19] (Fig. 2d). The expression of these markers was comparable in the two conditions and the presence of Ki67 in the slow-frozen shave biopsies revealed that cells can still proliferate after thawing. Ex vivo cultures of different types of cancer including melanoma, BCC, and breast cancer have been successfully used for many applications including drug testing[18,20]. Here, we show that it is possible to slow-freeze the tissue and use it later as ex vivo material without disturbing tumor morphology and proliferation. We thus assume that slow-frozen tissue can be used subsequent to storage for drug testing.

**Single-cell RNA sequencing uncovers similar degrees of cell-type heterogeneity in fresh and slow-frozen samples.** In order to compare scRNAseq in paired fresh and slow-frozen tumors from the same tissue, we sequenced over 100,000 cells from three BCC, two CRC, and one melanoma samples. The slow-frozen samples yielded similar numbers of cells as the fresh and the depth of the sequencing was comparable (Supplementary Fig. 2a). Cell types were called using the SingleR R package, cell-type calling was adjusted based on clustering and cell-type markers to identify

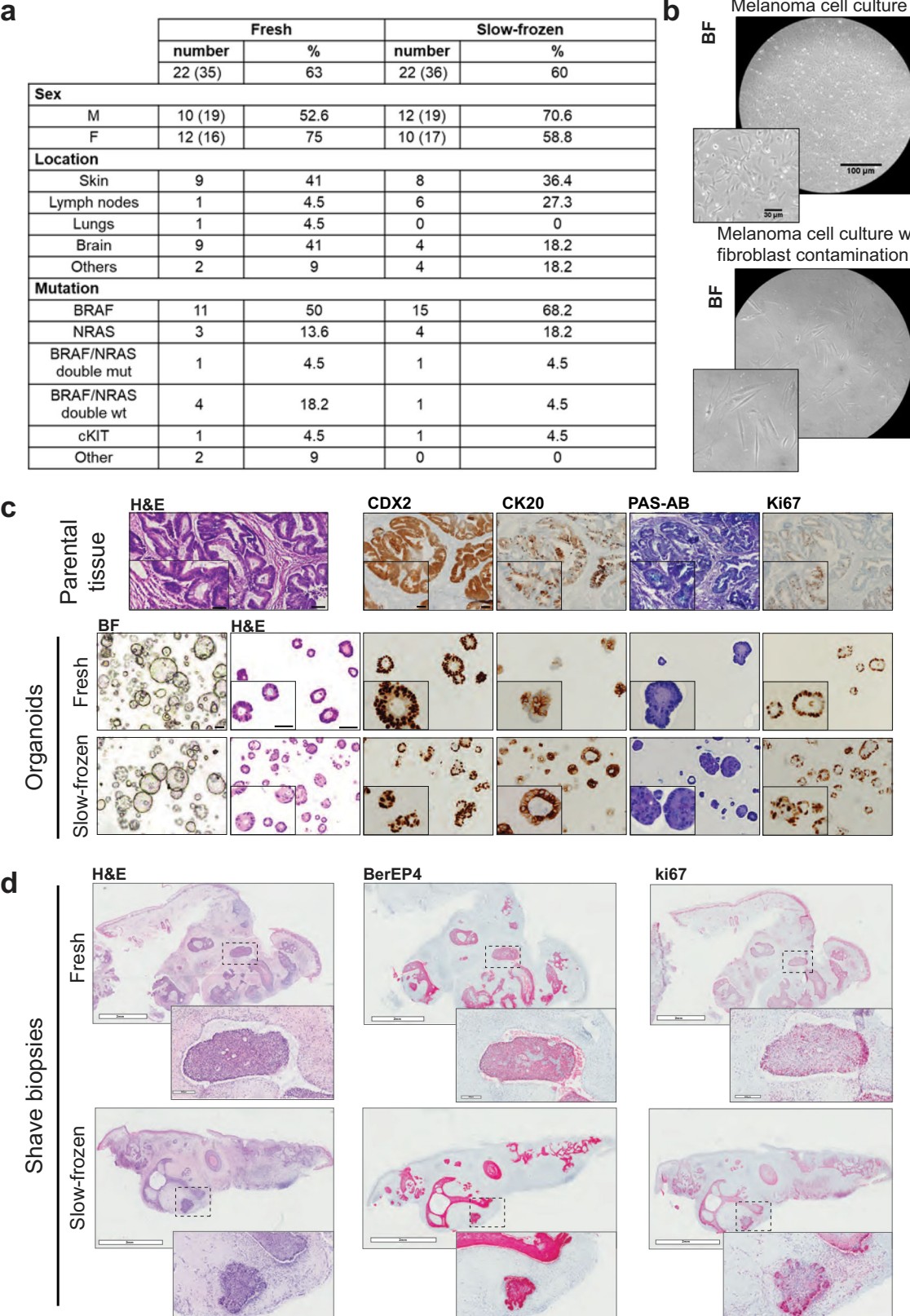

**Fig. 2 2D, 3D, and ex vivo culture from fresh and slow-frozen matching tumor samples. a** Percentage of successfully established melanoma cell lines from fresh or slow-frozen biopsies. **b** Representative bright-field micrographs of a successful melanoma cell line and a cell line contaminated with non-melanoma cells. Scale Bar 100 μm and 30 μm (insert). **c** Microscopic photographs of matched colon carcinoma primary parental tumor tissue and organoids derived from fresh and slow-frozen tissues. Scale bar lengths are 100 μm, for insert microscopic photographs 50 μm. CK20: Keratin 20, PAS-AB: Periodic acid-Schiff Alcian Blue, Ki67. **d** Microscopic photographs of shave biopsies from matched fresh or slow-frozen BCC stained with Hematoxylin-eosin (H&E) BErEp4 and Ki67 antibodies. Scale bar lengths are 2 mm and 200 μm.

Schwann-cells, mast cells, lymphatic and vascular endothelial cells, cancer-associated fibroblasts (CAFs), and to differentiate normal epithelial cells from tumor cells (Supplementary Fig. 2b). Cell-type composition plots of paired fresh and slow-frozen samples derived from the three cancer types showed that heterogeneity was preserved in slow-frozen pairs, but the relative proportion differed between fresh and slow-frozen samples (Fig. 3a). CAFs and vascular endothelial cells were enriched, and epithelial cells were underrepresented in the slow-frozen pairs in BCC but not in CRC samples. This suggests that the keratinocytes of BCC are more sensitive to slow-freezing, compared to the same cell population in CRC. Among immune cell populations, granulocytes such as mast cells and neutrophils were hardly detectable in the slow-frozen tissues, which highlights the importance of using fresh material when studying these cell populations. From all analyzed samples, the melanoma FNA's cell-type composition was changed the least by the slow-freezing procedure (Fig. 3a). Except for granulocytes, cell lineage markers were equally well detected in the majority of the cell types in fresh and slow-frozen samples (Fig. 3b). Uniform manifold approximation and projection (UMAP) visualization of six fresh and slow-frozen sample pairs (12 samples) showed a clear separation by cell type (Fig. 3c) and not by preservation method or sample origin (Fig. 4a) indicating that the effects of the slow-freezing and the enzymatic digestion treatment were not strong enough to obfuscate the differences between cell types. In order to validate our findings on a different cohort, we have performed the same analysis on the publicly available dataset containing paired fresh and frozen samples from breast cancer, prostate cancer, and melanoma tissue biopsies[9]. In these samples, we also observed that the cell-type heterogeneity of the samples was preserved after freezing, with only slight changes to the proportion of cell types. Importantly, we only detected neutrophil granulocytes in fresh samples and the proportion of mast cells decreased in the frozen samples (Supplementary Fig. 3a). Cells clustered together according to cell type and not according to treatment (Supplementary Fig. 3b, c).

**Cell-type dependent activation of heat-stress response in slow-freezing.** While cells from fresh and slow-frozen samples clustered together, fresh and slow-frozen samples separated more from each other depending on specific cell types (Fig. 4a). In order to quantify this, we have examined the correlation between the gene expression of fresh and frozen cells in each cell type. We observed high correlations between the gene expression of fresh and frozen cells in all cell types (Supplementary Fig. 4a). To put the correlation values into perspective, we have also calculated the correlation of gene expression between cells from the fresh samples of different patients (suffering from the same cancer type). On average, the correlation values between samples were lower than within the same sample comparing fresh and frozen cells (Fig. 4b). The correlations between the gene expression of fresh and frozen CAFs, fibroblasts, BCC cells, endothelial cells, and myeloid cells were lower than the correlations of these cells compared between samples of different patients (Fig. 4b and Supplementary Fig. 4b). Cells in tissues from BCC patients were affected more by freezing than in tissues from melanoma or CRC patients. To assess how these cell types change as a result of slow-freezing, we performed differential gene expression analysis between fresh and slow-frozen samples for each cell type. We have observed differential expression of the FOS and heat-shock genes in over 10 out of the 17 cell types (Fig. 4c). Among the Reactome pathways[21], most differentially regulated upon slow-freezing in most cell types were HSF1 activation and cellular response to heat stress (Supplementary Data). Cellular response to heat stress was most pronounced in CAFs, fibroblasts,

endothelial and myeloid cells (Fig. 4d), which were the cell types that were found most dissimilar between the fresh and slow-frozen samples. Despite the upregulation of those stress response pathways in these cells, the transcriptome of cells recovered from freshly processed and slow-frozen biopsies allowed the identification of cell types and expression of cell lineage markers (Fig. 3b).

**Single-cell RNA sequencing of slow-frozen breast cancer samples confirms the presence of clinically important markers.** To further demonstrate the strength of applying scRNAseq to live-biobank slow-frozen samples, we processed five primary breast cancer samples with various receptor statuses (Table 2). Resected breast cancer samples were split into two parts, one part was slow-frozen and the other part was preserved as an FFPE block (Supplementary Fig. 5a). The slow-frozen samples were thawed, enzymatically digested, and processed for single-cell profiling (Supplementary Fig. 5a). Sections of the matched FFPE tissues were prepared, stained for cell lineage and breast cancer-specific markers, scanned, and analyzed using the HALO image analysis platform (Supplementary Fig. 5a, b, c).

scRNAseq identified various cell populations within the tumor and its microenvironment, such as endothelial cells, keratinocytes, fibroblasts, CD4+/CD8+ T cells, myeloid cells, and epithelial cells (Fig. 5a). We used the dimensionality reduction method UMAP to control for batch effect (Fig. 5b). scRNAseq analysis for the receptor status of the breast cancer samples clinical subtypes was in alignment with the protein expression identified by IHC and reviewed by pathologists (Supplementary Fig. 6). Distinct gene expression signatures were observed in the tumors with different receptor profiles, with triple-negative breast cancer expression patterns being more similar to each other than to progesterone- and estrogen-receptor expressing tumors (Fig. 5c). Expression of cell type markers has been compared between Immunohistochemistry (IHC) on FFPE sections and scRNAseq of matched slow-frozen tissues. Gene expression and protein abundance for EpCAM/EpCAM, ESR1/ER, PGR/PGR, GATA3/GATA3 were detected in cancer epithelial cells. T-cell populations were detected by expression and abundance of CD3/CD3 and CD8/CD8 using both methods (Fig. 6a, b). In contrast, only CD4 and CD68 proteins have been detected (Fig. 6a, b). MKI67/Ki67 was low on both mRNA and protein levels. This result was expected since Ki67 was quantified by the pathologist with 15% on the original patient tissue (Fig. 6a, b, Table 2). Using the HALO image analysis platform, we quantified protein abundance and correlated it to the corresponding mRNA levels in scRNAseq (Supplementary Fig. 5d). Expression of mRNA and protein abundance in stromal and tumor cells showed a good correlation ($r = 0.62$, $p = 7.9^{-6}$) (Supplementary Fig. 5e), with a better correlation of mRNA expression and protein abundance in tumor cells only ($r = 0.68$, $p = 4.7^{-4}$) (Fig. 6c). Granulocytes have been shown to be sensitive to slow-freezing[22] and as expected were not recovered in slow-frozen samples (Fig. 3b). Notably, IHC staining of the paired FFPE section detected the neutrophil granulocyte marker myeloperoxidase (MPO), suggesting that FFPE material could recapitulate fresh material for proteome analysis (Fig. 6d).

## Discussion

Live preservation of biological material has advanced during the last decades. Several studies have been published describing applications of live freezing for various sample types (PDX, hair follicle, and tumors) and a wide range of applications[2–7]. In this work, we demonstrate that small pieces of tumors, FNAs, and shave biopsies derived from surplus surgery material, slow-frozen

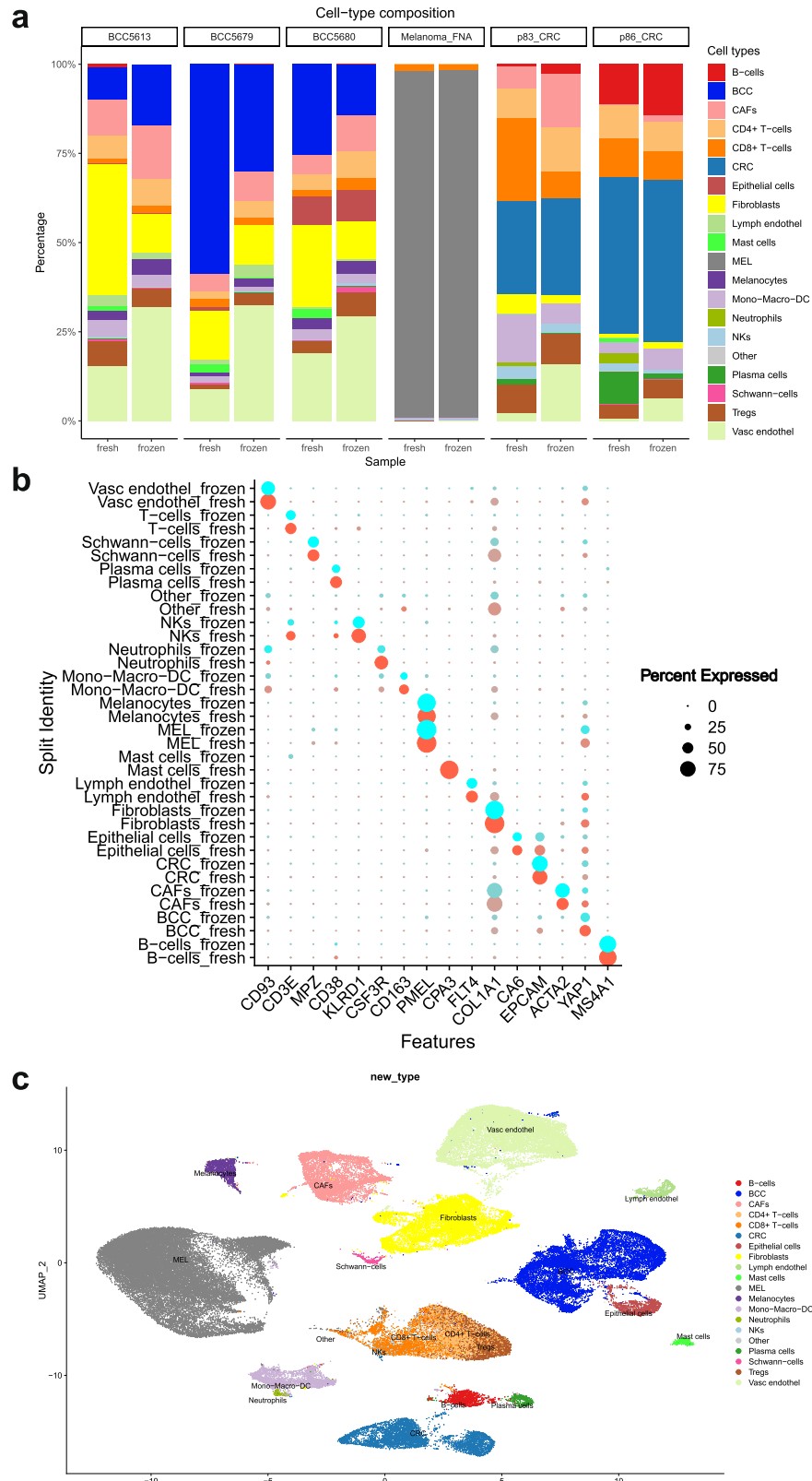

**Fig. 3 scRNAseq analysis of matching fresh and slow-frozen tumor samples. a** Stacked barplot showing the cell-type composition of matching fresh and slow-frozen tumor samples. **b** Dotplot showing the gene expression of cell lineage markers in major cell types in fresh and slow-frozen tumor samples. **c** UMAP plot of the matching fresh and slow-frozen tumor samples.

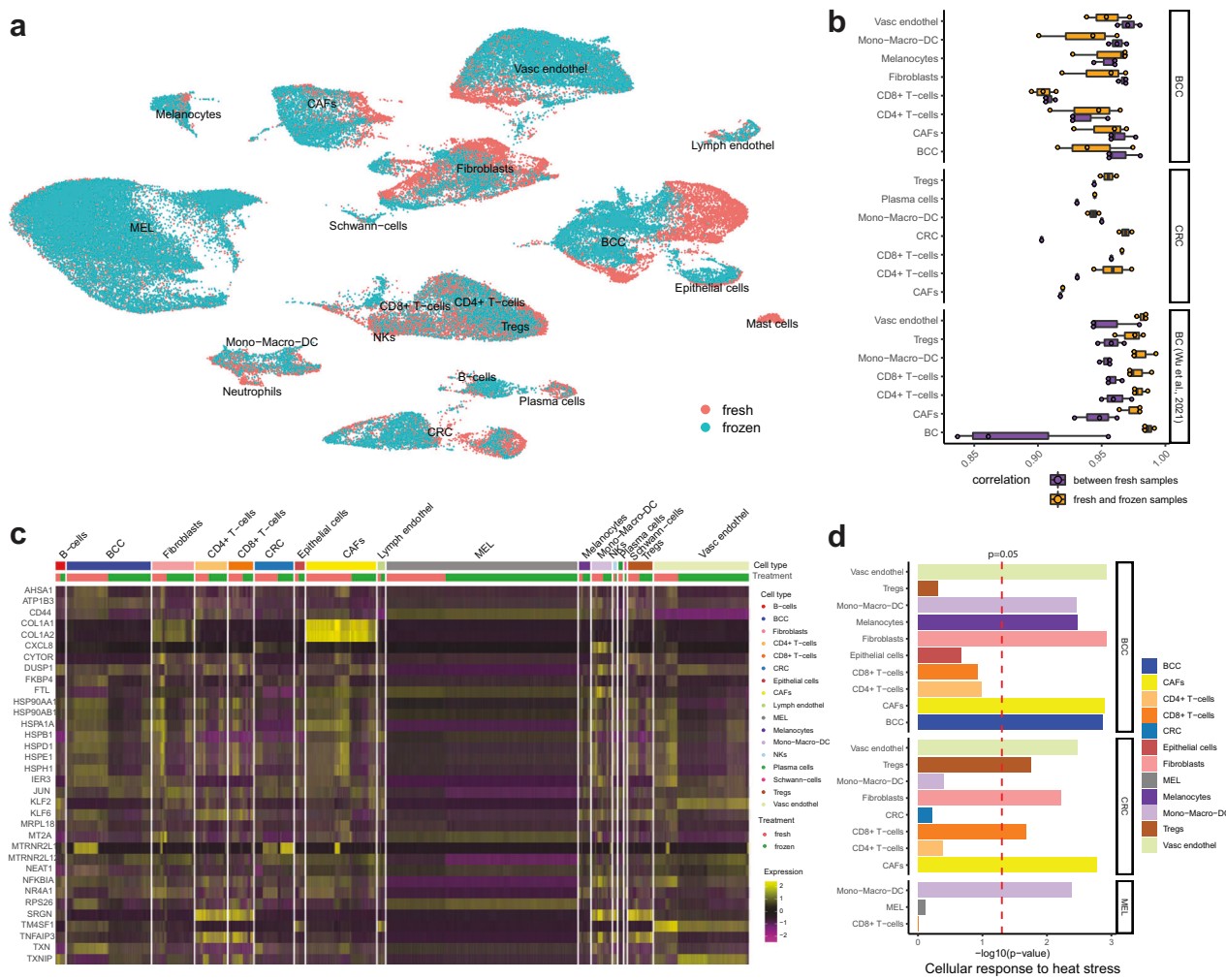

**Fig. 4 Correlation and differential gene expression analysis of fresh and slow-frozen scRNAseq tumor samples. a** UMAP visualization of fresh and slow-frozen samples according to the preservation method. **b** Boxplot showing the correlations between the gene expression of cells in fresh samples of different patients (purple) and the correlations between the gene expression of cells of fresh vs frozen samples (orange). Dots depict the correlation value for each sample pairing (e.g., Sample1_fresh vs Sample2_fresh or Sample1_fresh vs Sample1_frozen). **c** Heatmap showing differentially expressed genes between fresh and frozen samples (genes differentially expressed in at least 9 out of the 18 cell types). **d** Negative log10 *p* values of the enrichment analysis for the Reactome pathway "Cellular response to heat stress" for each cell type.

**Table 2 Slow-frozen cohort of breast cancer samples for scRNAseq and IHC image analysis.**

| Cancer type | Sample ID | Sample type | Diagnose | Age | Sex | Location | Application |
|---|---|---|---|---|---|---|---|
| Breast cancer | UHB150 | Resection | TNBC (ER-/PR-/HER2-) | 27 | F | Breast | scRNAseq (only frozen) |
| Breast cancer | UHB194 | Resection | TNBC (ER-/PR-/HER2-) | 64 | F | Breast | scRNAseq (only frozen) |
| Breast cancer | UHB129 | Resection | ER 100%, PR 70%, HER2- | 61 | F | Breast | scRNAseq (only frozen) |
| Breast cancer | UHB173 | Resection | ER 80%, PR 80%, HER2- | 83 | F | Breast | scRNAseq (only frozen) |
| Breast cancer | UHB182 | Resection | ER: 100%, PR: 100%, HER2 - | 90 | F | Breast | scRNAseq (only frozen) |

in FCS with 10% DMSO shortly after collection can be used for different applications. In particular, we show that 2D, 3D, and ex vivo culture models can be established from small tumors and shave biopsies and that the success rates and features are maintained when compared to the corresponding fresh material (i.e., used directly after collection, without slow freezing). The feasibility of 2D and 3D models has also been described by He et al.[15] for different types of cancer. Moreover, in accordance with another study[9], we have shown that scRNAseq can be applied to slow-frozen small tumors and that the overall tumor

heterogeneity and microenvironment are maintained with different proportions of certain cell types and minimal transcriptome differences between fresh and slow-frozen tissue pairs. Similarly to Wu et al.[9], we also found that slow-freezing affected mainly cold response genes such as those belonging to the jun-fos pathway and heat-shock proteins, albeit to a different degree, depending on the cell type. Denisenko et al.[23,24] have confirmed this observation in adult mouse kidneys and noted that cryopreservation specifically depleted epithelial cells of the proximal tubuli. In this study, we observed that epithelial cells (including

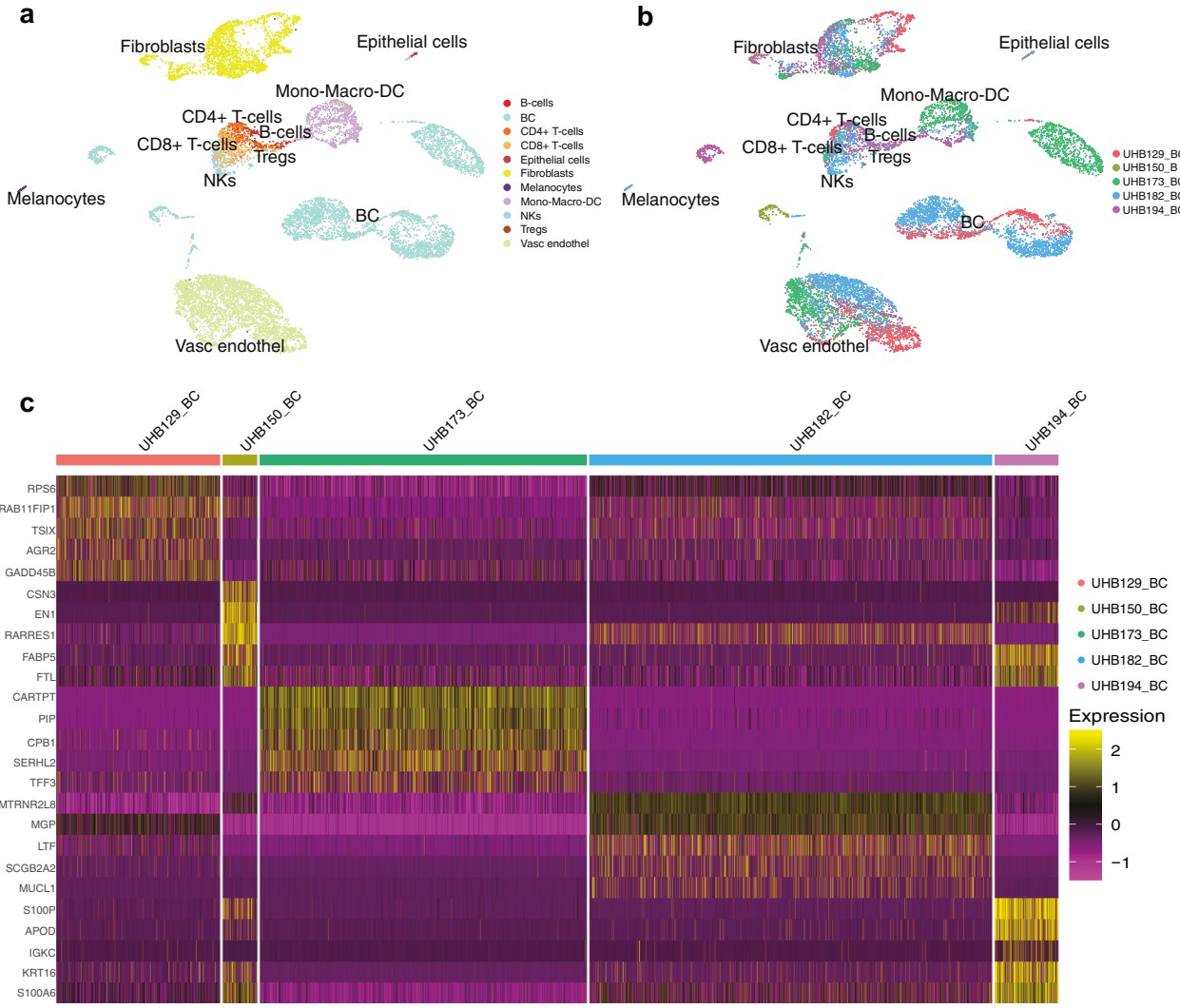

**Fig. 5 scRNA sequencing of slow-frozen breast cancer samples. a** UMAP plot showing major cell populations in five breast cancer samples. **b** UMAP visualization of five breast cancer samples by clinical subtypes. **c** Heatmap showing the five most differentially expressed genes in the breast cancer samples from different clinical subtypes: UHB129/UHB173/UHB182 are hormone receptor-positive and UHB150/UHB194 are triple-negative subtypes.

keratinocytes) and CAFs were affected more in slow-frozen BCC cancer cells compared to slow-frozen CRC which could be due to shorter enzymatic digestion times or tissue-specific characteristics. This was even clearer in the melanoma FNA, where no digestion procedure was applied and the fewest differences between the fresh and the slow-frozen samples were detected. In line with previous results[22], our analysis showed that among immune cells most affected types were neutrophils and mast cells, which were almost completely absent from slow-frozen samples. Furthermore, we have shown in slow-frozen breast cancer samples that scRNAseq can detect important clinically relevant markers and validate their presence with conventional IHC at the protein level.

One limitation of our study is that we have not tested if slow-freezing procedure have any further effect on the tissue quality and could affect dwonstream application in samples stored longer than 14 months. However, we can hypothesize that slow-frozen tumor pieces are well preserved similar to cell lines stored for years in liquid nitrogen, provided that tissue pieces are small enough (maximum size 2 × 2mm) to allow DMSO penetration and stable storage temperature are guaranteed. Based on our results we can conclude that tumor pieces can be stored for up to 1 year as slow-frozen samples for future downstream analysis.

In conclusion, this work opens up a broad spectrum of applications for surplus clinical material from different solid cancers that cannot be used immediately after collection.

## Methods

**Tumor specimen collection and live biobanking**. Tumor resections and biopsies, including shave biopsies, were collected directly after surgery from consenting patients at different hospitals in Switzerland. All samples used were surplus materials from routine surgeries. BCC biopsies were provided by the Dermatology Department of the University Hospital Zürich with the assistance of the SKIN-TEGRITY.CH biobank. CRC resections were provided by the University Hospital Basel and St. Clara Hospital Basel and breast cancer biopsies were provided by the University Hospital Basel. Informed consent had been obtained from all patients and all experiments conformed to the principles set out in the WMA Declaration of Helsinki and the Department of Health and Human Services Belmont Report. The use of material for research purposes was approved by the corresponding cantonal ethic commissions (Zürich (Biobank): EK-687 and 800, (BCC) KEK 2017-00688, Basel: (CRC) EKBB 2019-00816, (Breast Cancer) EKNZ 2018-00729).

Once collected, the tissues were divided in two; one part was freshly processed (BCC and CRC) for different applications and the other part was cut into small pieces of 2–3 mm with a sterile scalpel blade and cryopreserved in 1.5 ml of freezing medium (90% FCS and 10% DMSO) in Mr. Frosty cooling container (cat. no. 5100-0001, NALGENE™ Cryo), and immediately transferred to −80 °C. After overnight cooling, the slow-frozen samples were transferred to liquid nitrogen (for longer storage periods). Breast cancer tissues were split in two and either cut into 3 mm³ pieces followed by slow-freezing in Bambanker freezing medium (Cat. No. BB01; NIPPON Genetics) and stored at −80 °C for single-cell analysis or fixed in

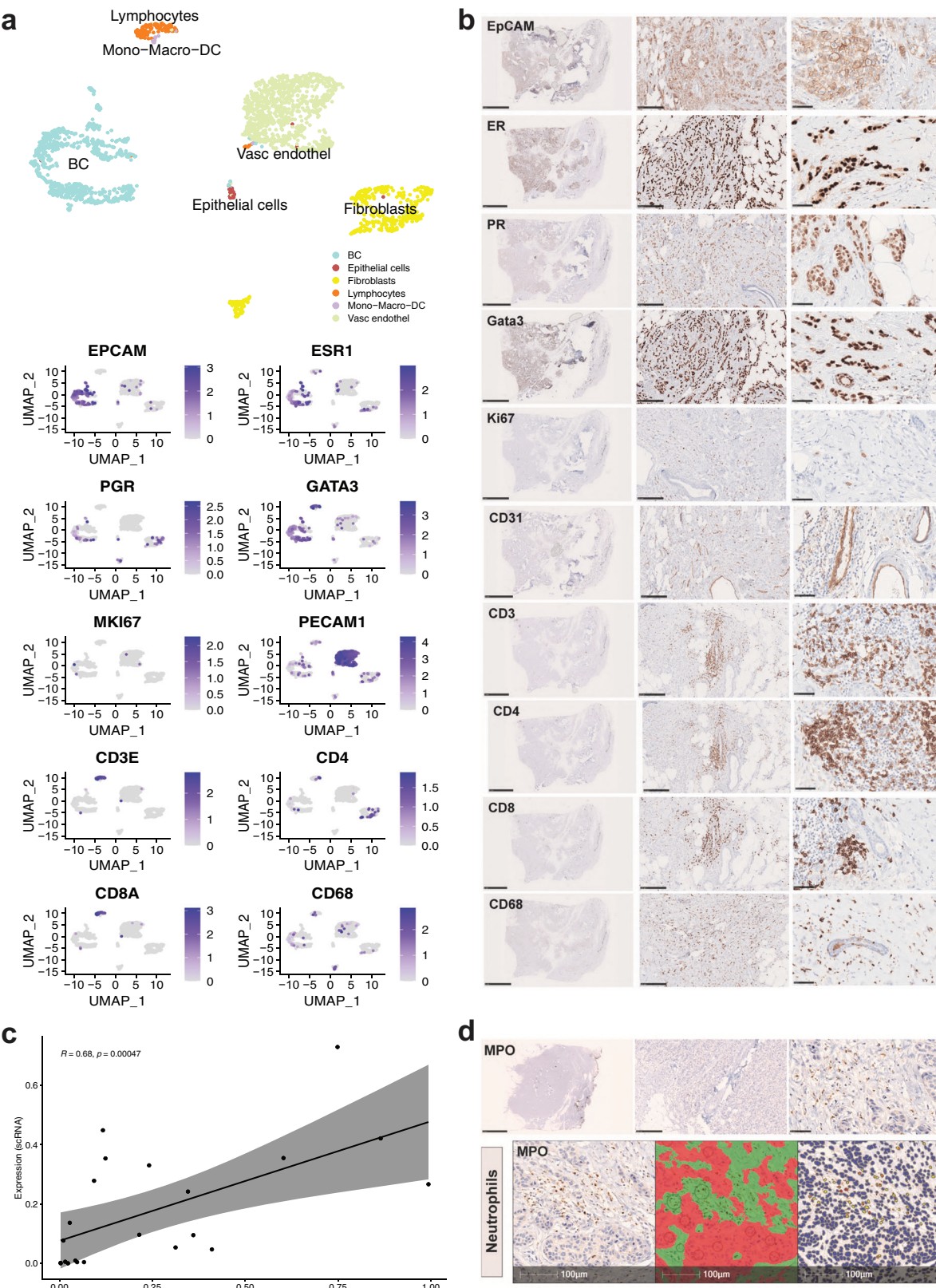

**Fig. 6 Correlation analysis of genes (from scRNAseq analysis of slow-frozen samples) and proteins (from immunohistochemistry stainings of FFPE sections) expression in breast cancer. a** UMAP visualization of major cell types in UHB129 (hormone receptor positive) breast cancer tissue sample and marker expression on transcriptome level. **b** IHC images showing corresponding protein expression; scale bar 5 mm, 250 μm, 50 μm. **c** Correlation of mRNA and protein expression in tumor cells. **d** Top: Representative IHC staining for neutrophil granulocytes marker, Myeloperoxidase (MPO) on FFPE section of UHB129 (hormone receptor positive) breast cancer tissue, scale bar: 5 mm, 250 μm, 50 μm. Bottom: Tissue classification (red: tumor cells, green: stroma) and cell, MPO staining quantification using HALO imaging analysis software, scale bar: 100 μm.

formalin and embedded in paraffin for subsequent immunohistochemistry analysis. For scRNAseq, cell isolation, and organoid preparation, slow-frozen tumor pieces were thawed at 37 °C in a water bath and dissociated with the optimal enzymatic digestion protocol for the corresponding technique (see scRNAseq chapter below). FNA was collected as described previously[25]. The melanoma FNA specimen was treated as the biopsies, half of the aspirate was processed for scRNAseq and half slow-frozen and subsequently processed.

**2D and 3D cell cultures.** 2D and 3D cell cultures were established from surplus material obtained from routine surgical removal of tumors. Only tissue from patients who had signed an informed consent approved by the local institutional research board was used for further applications.

**Melanoma cell line isolation and culturing.** Surplus tumor material was obtained after surgical removal of melanoma metastases. Clinical diagnosis of the tumor material was confirmed by histology and immunohistochemistry. Moreover, mutations of the tumors were assessed routinely with the MelArray platform or Sanger sequencing (genes on the panel of MelArray are in Supplementary Table 3). Primary melanoma cells were established from fresh tissue or slow-frozen tissue (not paired) using the selective adherence method[26]. Briefly, tumor material was divided into small pieces and digested with 2.4 U/ml dispase (Roche, Basel, Switzerland), followed by digestion with 62.5 U/ml collagenase (Sigma, St. Louis, MO, USA), so that the final suspension consisted of separated cells. This suspension was cultured in RPMI1640 (Invitrogen, Carlsbad, CA, USA) supplemented with 5 mm L-glutamine (Biochrom, Berlin, Germany), 1 mm sodium pyruvate (Gibco, Carlsbad, CA, USA), and 10% FCS (Gibco) at 37 °C and 5% $CO_2$. Non-melanoma cell contamination (e.g., fibroblasts) was avoided by differential trypsinization or by inducing fibroblast senescence through contact inhibition, serum starvation, or deprivation of adhesion. Those conditions allowed in most cases pure melanoma primary cultures. Cultures were split when they were approximately 90% confluent, and the medium was refreshed twice per week. To confirm the mutation found in the parental tumor, DNA from cells was isolated and amplified with primers for the specific genes mutated in the parental tumor (Supplementary Table 4). For DNA isolation we used QIAamp DNA MiniKit (Qiagen, Cat. No.51304) and for amplification, we used the AmpliTaq Gold kit (Roche, Cat. No. N8080241). After amplification, we sequenced the material using the BigDye™ Terminator v1.1 Cycle Sequencing Kit (Applied Biosystems™, Cat. No. 4337452). The sequencing was performed using the M13 primers or primers specific for the sequence to amplify and with the 3500 genetic analyzer (Applied Biosystem, Cat. No. 4405673). The amplified products were aligned to the wild-type sequences with the BLAST function of NCBI.

**Generation of CRC patient-derived organoids.** Surgically resected tissues were transported to the laboratory (Department of Pathology Basel) on ice-cold MACS buffer (Miltenyi, Cat. No. 30-100-008) or on ice-cold DMEM medium (Gibco, Cat. No. 41966052). Colon and rectum are microbiota-containing organs[27] therefore, have an implicit risk of microbiota contamination for the PDO culture. To avoid bacterial contamination, CRC primary tissues were washed three times on ice for 5 min each with the following washing solution: PBS-Primocin (0.1 mg/ml, Invivogen cat NC9141851) and Penicillin/Streptomycin (20 Unit/ml/20 µg/ml, GIBCO, Cat. No. 10010023/cat GIBCO, Cat. No. 10378016). Then, both CRC primary and metastasis tissues were cut in two equal parts. One part was cryopreserved with FCS with 10% DMSO in order to preserve cell viability and was stored from 1 to 4 weeks; the other part was processed immediately to generate PDOs as previously described (ref: 10.7150/thno.50051). Briefly, the tissues were cut into small pieces and digested in 5 mL advanced DMEM/F-12 (GIBCO, Cat. No. 12634028) containing 2.5 mg/mL collagenase IV (Worthington, Cat. No. LS004189), 20 µg/mL hyaluronidase V (Sigma, Cat. No. H6254), 1% BSA (Sigma, Cat. No. A3059), 0.1 mg/mL DNase IV (Sigma, Cat. No. D5025) and 10 µM LY27632 (Abmole Bioscience, Cat. No. M1817) for ~1 hour and 30 min at 37 °C under slow rotation and vigorous pipetting every 15 min. The tissue lysate was filtered through a 100 µM cell strainer, centrifuged at 300 RCF for 15 min and then treated with Accutase (Sigma, Cat. No. A6964) for 10 min at room temperature to dissociate the remaining fragments. After 10 min centrifugation at 300 RCF, the cell pellet was suspended in PBS and cells were counted using trypan blue (GIBCO, Cat. No. 15250061), Countess™ Cell Counting Chamber Slides (Invitrogen, Cat. No. C10228) in a Countess™ II FL Automated Cell Counter (Invitrogen, Cat. No. AMQAF1000). Then, the cells were mixed with growth factor reduced Matrigel (Corning, Cat. No. 356231) and seeded as drops in a tissue-culture dish. After polymerization of Matrigel, culture medium supplemented with growth factors was added to the cells. The composition is advanced DMEM/F-12 supplemented with penicillin/streptomycin, 10 mM HEPES (GIBCO, Cat. No. 15630056), 2 mM GlutaMAX (GIBCO, Cat. No. 35050038), 500 ng/ml R-Spondin (Pepro-Tech, Cat. No. 120-38) 100 ng/ml Noggin (PeproTech, Cat. No. 120-10 C), 1× B27 (Life Technologies, Cat. No. 17504-044), 1.25 mM n-Acetyl Cysteine (Sigma, Cat. No. A9165), 10 mM Nicotinamide (Sigma, Cat. No N0636), 50 ng/ml human EGF (PeproTech, Cat. No. AF-100-15), 10 nM [Leu15]-Gastrin I human (Sigma, Cat. No G9145), 500 nM A83-01 (Tocris Bioscience, Cat. No. 2939), 3 µM SB202190 (Sigma, Cat. No. S7076), 10 nM Prostaglandine E2

(Tocris Bioscience, Cat. No. 2296), 10 µM LY27632 and 100 µg/ml Primocin. The medium was changed every 3 days, and organoids were passaged after dissociation with 0.25% Trypsin-EDTA (GIBCO, Cat. No. 25200056).

**Immunohistochemistry of organoids.** PDOs were released from Matrigel by incubation with Dispase (1 mg/ml, Thermo Fisher, Cat. No.17105-041) for 45 min. Organoids were then fixed in freshly prepared 10% Paraformaldehyde (PFA) solution in PBS for 30 min at room temperature following dehydration and paraffin embedding. Sections were stained with H&E, Alcian blue-periodic acid-Schiff (PAS) as well as immunohistochemical staining, according to standard protocols. Histopathological evaluation was assessed by a board certificated pathologist (C.E.). Tumors were classified based on architecture and cytological features. For immunohistochemistry, the following primary antibodies were used for automated staining on a VENTANA BenchMark Ultra immunohistochemistry staining system (Bond, Leica): KI67 (Dako Cat. No. IR626), CDX2 (Ventana Cat. No. 760-4380) and CK20 (Ventana Cat. No. 790-4431).

**Culturing of BCC shave biopsies.** BCC shave biopsies were obtained from the dermatology department of the university hospital Zürich (USZ). Only material from consenting patients was used for this purpose. After collection, one shave biopsy was cut into 2 pieces, 1 piece was put directly in cell culture insert (Millipore, Cat. No. Z352985), and the other part slow-frozen. The insert was placed in a 6-well plate and the shave biopsy on the insert was cultivated for 5 days with 1.5 ml of co-culture medium only at the bottom. The medium was changed every second day. After 5 days, IHC was performed to measure proliferation (Ki67) and the presence of BCC cells (BerEP4). The co-culture medium is composed as follows: 3 parts DMEM (Gibco, Cat. No. 41966052), 1 part Ham's F-12 Nutrient Mix (Gibco, Cat. No. 11765054), 10% Fetal Bovine serum heat inactivated (Gibco, Cat. No.1050), 0.1 mg/ml of Normocin™ (Invivogen, Cat. No. ant-nr-1), 21.8 µg/mL of Adenin (Sigma-Aldrich, Cat. No. A2786), 5.45 µg/mL of Apotransferrin (Sigma-Aldrich, Cat. No. T1147), 2.18 nM of Triiodothyronine (Sigma-Aldrich, Cat. No. T6397), 0.44 µg/mL of Hydrocortisone (Sigma-Aldrich, Cat. No. H0888), 0.11 nM of Cholera toxin (Sigma-Aldrich, Cat. No. C8052), 5.50 µg/mL of Insulin (Sigma-Aldrich, Cat. No. I6634) and 0.01 µg/mL of Epidermal growth factor (Sigma-Aldrich, Cat. No. E4127)

**IHC of shave biopsies.** Blocks of paraffin-embedded, formalin-fixed tissues from BCC shave biopsies were used for IHC. Briefly, shave biopsies were fixed overnight in 4% formaldehyde then dehydrated and put in paraffin. 4um cuts were produced and were used to perform IHC. Sections were stained with H&E. For immuno-histochemistry, the following primary antibodies were used for automated staining on a VENTANA BenchMark Ultra immunohistochemistry staining system (Roche): Ki67 (DAKO, M7240, 1:50) and BerEP4 (Dako. M0804, 1:300). Images were scanned with Aperio ImageScope Slide Scanner.

**IHC and analysis of breast cancer biopsies.** Breast cancer tissues were obtained from tumorectomies or mastectomies performed at the University Hospital Basel. All participating patients have signed an informed consent approved by the responsible authority (EKNZ Ethical approval project ID: 2018-00729) allowing the usage of their tissues for scientific research purposes. Tumor tissues were transferred to the laboratory (Department of Biomedicine, Department of Surgery, University Hospital Basel, University of Basel, Switzerland). Breast cancer tissues were split in two and either cut in 3 mm³ pieces followed by slow-freezing in Bambanker freezing medium (Cat. No. BB01; NIPPON Genetics) and stored at −80 °C for scRNAseq or fixed in formalin and embedded in paraffin for subsequent immunohistochemistry analysis. For immunohis-tochemistry breast cancer tissues were fixed in 4% PFA followed by dehydration, paraffin embedding, and sectioning (4 µm). Immunohistochemistry was performed on the BenchMark ULTRA IHC/ISH system (Ventana/Roche) by the Department of Pathology, University Hospital Basel using the antibodies listed in Table 3.

IHC images were acquired with the NanoZoomer S60 Digital slide scanner (Hamamatsu) and images were analyzed with the HALO imaging analysis platform (v3.1; IndicaLabs) applying the Random Forest classifier (based on tissue marker expression or tissue structure) and the Multiplex IHC 2.3.4 algorithm.

**Single-cell RNA sequencing.** Fresh samples were processed within 2 h after sampling (FNA of metastatic melanoma) or after overnight storage in a complete medium (RPMI, Cat. No. R0883, Sigma) with 10% FCS (Biowest, Cat. No. S006420E01, batch no. S169419181H), 1% Pen-Strep (Gibco, Cat. No. 15070063) at 4 degrees (basal cell carcinoma and primary colorectal cancer). Cryopreserved tissue samples were quickly thawed in the water bath set to 37 °C degree, re-suspended in 10 mL of ice-cold RPMI with 0.04% BSA (Sigma-Aldrich, Cat. No. A7906), and incubated for 10 min on ice to allow DMSO release from the tissue. After samples were spun down at $300 \times g$ for 5 min, cut into small pieces, and enzymatically digested for 30 min–2 h min at 37 °C on a shaker using appropriate digestion protocol:

**Table 3 Antibodies used to stain the breast cancer tissues and corresponding dilutions.**

| Antibody | Company and Cat. No. | Dilution |
|---|---|---|
| CD11c | Cellmarque, 111M-18 | RTU |
| CD14 | Ventana, 760–4523 | RTU |
| CD19 | Dako, IR656 | RTU |
| CD3 | Ventana, 790–4341 | RTU |
| CD31 | Ventana, 760–4378 | RTU |
| CD4 | Ventana, 790–4423 | RTU |
| EpCAM | Ventana, 760–4383 | RTU |
| ER | Ventana, 790–4324 | RTU |
| FoxP3 | Abcam, ab99963 | 1:50 |
| Gata3 | Ventana, 760–4897 | RTU |
| SMA | Ventana, 760–2833 | RTU |
| Sox10 | Ventana, 760–4968 | RTU |
| CD56 | Ventana, 790–4465 | RTU |
| CD68 | Dako, IR613 | RTU |
| CD8 | Ventana, 790–4460 | RTU |
| CK14 | Ventana, 760–4805 | RTU |
| CK22 | Immuno, Bio MM-1012-02 | 1:2000 |
| CK5/6 | Ventana, 790–4554 | RTU |
| CK8 | Ventana, 760–2637 | RTU |
| HER2 | Ventana, 790–2991 | RTU |
| Ki67 | Dako, IR626 | RTU |
| PR | Ventana, 790–2223 | RTU |
| Vimentin | Ventana, 790–2917 | RTU |

*RTU* ready to use, antibody-working concentration is used according to manufacturer's instructions.

| Tissue type | Digestion protocol |
|---|---|
| BCC | Sequential digestion protocol:<br>Step 1: 2.4 U Dispase II (Roche, Cat. No. 04942078001) − 1–2 h at 37 °C<br>Step 2: 1000U Collagenase IV (Worthington, LS004188) + 15 KU DNAse (Sigma, Cat. No. D5025) - 30 min at 37 °C<br>Step 3: Trypsin 0.25% (Gibco, Cat. No. 25200-56)- 10 min at 37 °C |
| CRC | One step digestion protocol:<br>5000U Collagenase IV (Worthington, LS004188),<br>15 KU DNAse I (Sigma, Cat. No. D5025),<br>2 mL Accutase (Sigma, Cat. No. A6964)<br>10 μM LY27632 (Abmole Bioscience, Cat. No. M1817)<br>0.5% BSA (Cat. No. A7906, Sigma-Aldrich)<br>dissolved in 5 mL RPMI with 2 mM CaCl2 (Cat. No. 746495, Sigma-Aldrich)<br>Digestion up to 1 hour at 37 °C |
| BC | One step digestion protocol:<br>Liberases$^{DH}$ Research Grade (Roche, Cat. No. 05401054001) up to 2 h at 37 °C |

After incubation, digested tissue was filtered through 100 μm nylon (Falcon, Cat. No. 352360) and then through 35-μm cell strainers (blue capped FACS tubes Cat. No. 352235, Falcon). For samples with viability below 80% apoptotic and dead cells were removed using immunomagnetic cell separation with the Annexin Dead Cell Removal Kit (Cat. No. 17899, StemCell Technologies) and EasySep™ Magnet (Cat. No. 18000, StemCell Technologies). If the cell pellet appeared red, red blood cell lysis was performed following the manufacturer's instructions (Cat. No. 11814389001, ROCHE). Cell number and viability were assessed on Luna-FL™ Dual Fluorescence Cell counter (Logos Biosystems Inc.) using Photon slides (Ultra-low fluorescence counting slides, Cat. No. L12005, Logos Biosciences Inc.) by acridine orange propidium iodide stain (AOPI, Cat. No. F23001, Logos Biosciences Inc.) and optimal cell concentrations were set to 700-1100 cells/μL according to 10× Genomics protocols. Single-cell droplets were generated using a 10x Genomics Chromium Single Cell Controller (PN110211, 10X Genomics), Chromium Next GEM Single Cell 3′ Kit v3.1 profiling kit (PN-1000122, 10X Genomics) and Next Gem Chip G Single cell kit (Cat. No. PN1000120, 10X Genomics). cDNA traces were amplified and GEX libraries were constructed (Library construction kit Cat. No. PN1000157; Single Index Kit T, Set A, Cat. No. PN1000213, 10X Genomics) according to the manufacturer instructions. 6000 cells per sample were targeted and the quality of cDNA traces and constructed gene expression libraries were evaluated on Agilent 2100 Bioanalyzer (system no. G1030AX, Agilent Technologies) using High Sensitivity DNA kit (Cat. No. 5067-4626, Agilent Technologies). Sequencing strategy: libraries were diluted to 10 nM and pooled at balanced ratios according to the target cell number. Paired-end sequencing (PE 28/8/0/91) was performed on an Illumina NovaSeq S2 flow cell at the Genomics Facility Basel of the University Basel and the Department of Biosystems Science and Engineering, ETHZ and Functional genomics center Zurich. According to the recommendation of 10x Genomics, a coverage of 50,000 read pairs per cell was targeted.

**Single-cell RNA sequencing analysis**. Raw sequencing data were processed using the 10× Chromium Cellranger pipeline (version 3.1.0) (https://support.10xgenomics.com/single-cell-gene-expression/software/downloads/latest). Reads were aligned to the human reference genome (GRCh38-3.0.0) (10x Genomics). Seurat (version 4.0)[28,29] was used to merge, scale and normalize gene expression data, as well as for clustering, differential gene expression analysis and visualizations. We used SingleR (version 1.4.1)[30] with the Blueprint[31] and ENCODE[32] reference datasets to assign cells to cell types. SingleR cell typing was restricted to main cell types as detailed in supplementary table 5 cell-type merging. As Schwann-cells, mast cells, CAFs and cancer cells are not found in the above-mentioned databases, these cells were identified based on clustering and the expression of known cell-type markers. Raw data from the study of Wu et al.[9] was accessed at (https://singlecell.broadinstitute.org/single_cell/study/SCP1415/cryopreservation-of-human-cancers-conserves-tumour-heterogeneity-for-single-cell-multi-omics-analysis). When calculating correlations between the average gene expression of cell types in each sample and when searching for enriched pathways, only cells in G0/G1 phase were considered. Gene-set enrichment analysis[33] was performed using the fgsea (https://www.biorxiv.org/content/10.1101/060012v3) R package, the Reactome pathway database was accessed using the msigdbr R package (msigdbr: MSigDB Gene Sets for Multiple Organisms in a Tidy Data Format (r-project.org)). The R package ggplot2 (https://wires.onlinelibrary.wiley.com/doi/abs/10.1002/wics.147) was also used for visualizing aggregated data.

**Statistics and reproducibility**. Multiple patients' samples were used for colorectal cancer, breast cancer, and basal cell carcinoma to give statistical power to our investigations. The explorative nature of the study did not allow for estimating effect sizes prior to the analysis. Tissue samples were processed, divided into equal parts, and randomly assigned either to be processed fresh or after freezing. All the scRNAseq samples passed the QC of the 10X cell-ranger data analysis pipeline (https://support.10xgenomics.com/single-cell-gene-expression/software/overview/welcome).

**Reporting summary**. Further information on research design is available in the Nature Research Reporting Summary linked to this article.

## Data availability
The scRNAseq data generated in this study have been deposited to the European Genome-phenome (EGA) database under accession code EGAS00001005891. Raw data from the study of Wu et al.[9] was accessed at (https://singlecell.broadinstitute.org/single_cell/study/SCP1415/cryopreservation-of-human-cancers-conserves-tumour-heterogeneity-for-single-cell-multi-omics-analysis).

## Code availability
The code used for the analysis of the scRNAseq data is available at: https://github.com/uzh-dqbm-cmi/scRNA-slow_frozen.

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

## Acknowledgements

We would like to thank Federica Sella and Ramon Lang for technical support with staining, Ina Nissen, and Elodie Burcklen for technical support with sample sequencing. We thank Dr. Egle Ramelyte for the melanoma FNA. This work was supported by the Swiss Cancer Research foundation (KFS-4459-02-2018) and by the Swiss Personalized Health Network (SPHN) project "Swiss Personalized Oncology (SPO)" work package 5, 2017DRI21 (PHRT111). Moreover, the study was supported by the SKINTEGRITY.CH consortium and Monique Dornonville de la Cour—Stiftung.

## Author contributions

M.P.L., M.K., S.P. designed the study. G.R., A.T., F.P., M.D., B.T.P., C.B., S.M.S. performed the experiments. Z.B., G.R., A.T., M.D., B.T.P., C.E. analyzed the data. A.T., G.R., Z.B., F.P., M.D. prepared the figures. G.R., A.T., Z.B. wrote the manuscripts. J.H., W.P.W., C.K., and M.V. assisted with providing human samples, M.P.L., M.K. supervised the project. All authors contributed to the editing of the final manuscript.

## Competing interests

The authors declare no competing interests.
