## [Peer Review File · Communications Biology]

Reviewers' comments:

Reviewer #1 (Remarks to the Author):

Remarks to be sent to the author:

Overall comment:

The manuscript 'Live slow-frozen human tumor tissue viable for 2D, 3D, ex vivo cultures and scRNAseq' by Restivo et al., compares using fresh vs. slow-frozen tissues from various malignancies for establishing different methods such as 2D, 3D, organoids, ex vivo cultures as well as scRNAseq. The manuscript is well written and follows a logical order. It covers 3 different cancer types and explores different growth conditions including state-of-the-art organoids and scRNAseq. However, despite the authors examining the important aspect of freezing and reusing human tissues for various research applications, the manuscript lacks exploration of some important technical aspects such as the effects of tissue freezing times and the various digestion protocols used.

Additional specific comments:

1. The paper does not come across as being high on innovation. The authors themselves summarize previous work that has shown similar results to what they show. It is true that the authors go beyond and show similarities on the single-cell level while focusing on the immune subtype, however, their conclusions are not always biologically clear. For example, in page 7, line 203, the authors fail to explain what the results indicate: 'FOS and heat-shock proteins in over 10 out of the 17 cell types'. It is simply not clear what this conclusion means. The authors should highlight and emphasize more the novel and innovative aspects of their work, as this is currently missing.

2. As mentioned above, the authors do not discuss the effects of long-term freezing and whether it influences the quality of tissue, nor do they discuss digestion protocols. There have been papers in the past (e.g., Quintana, 2008) discussing how digestion protocols affects tumor formation and the number of tumor-initiating cells. Can the authors please comment and include this in their analyses? They mention this briefly in line 265, but this is not sufficient.

Minor comments:

1. There are some typos where the authors sometimes use μ to indicate micro, and sometimes they use u.
2. In p. 16, the table is missing a cat.no. It is simply written as xx.

Reviewer #2 (Remarks to the Author):

Restivo et al. describe a comparison of cryopreserved vs. freshly processed human tumour tissues for culture and scRNAseq, and report encouraging results indicating that cryo-preserved human tumour tissues (at least from colon cancer and basal cell carcinoma) remain representative of the original tumour material to a similar extent as freshly processed tumour tissue samples, as determined by establishing tumour tissue cultures and by performing scRNA sequencing.

After reporting comparable success rates for establishing tissue cultures from cryopreserved melanoma samples as for freshly processed ones, the authors then analyse scRNAseq data or tissue culture establishment in matched fresh vs. slow-frozen samples from each of 5 BCC (3 scRNAseq, 2 culture), 4 colon cancers (2 scRNAseq, 2 culture) and one melanoma fine needle aspirate (scRNAseq). I think that this section is the most novel part of the manuscript, and I find it particularly impressive that the authors analysed matched slow-frozen vs. fresh processed samples from each of the tumours.

In addition, the authors also studied five slow-frozen breast cancer cases by scRNAseq (i.e. without matching fresh processed material).

I was impressed by the thorough analysis of the scRNAseq in matched fresh vs. slow-frozen samples, showing similar cell-type heterogeneity, albeit with preferential loss of granulocytic cells in slow-frozen samples.

I think it would be a good idea for the authors to include a comparison of their work with that reported by He et al in 2020 (Biopreservation and Biobanking, Vol 18, No. 3 p. 222-227) DOI: 10.1089/bio.2019.0062. I think Restivo et al.'s work compares favourably with that of He et al, and it would be sensible to compare the approaches and findings in the two papers.

Some specific comments

Page 1 line 17: Who are the authors who contributed equally? I do not see any marked by the '7' in superscript

Page 3 line 65: 'showed' instead of 'have shown'

Page 3 line 65: define PDX at first use (patient-derived xenograft)

Page 4 line 94: perhaps 'broad' rather than 'broader'?

Page 4, line 110: 'compared slow-freezing methods': it seems that only one method (singular, not plural) for slow-freezing was used in this work; the comparison is with fresh processing of tumour tissue and the outcomes being tested are scRNAseq characteristics and establishment of tumour cultures

Page 13, line 402: Ventana Benchmark Ultra is made by Roche, not by Leica Bond (Leica Bond is a different platform for immunohistochemistry)

Page 23, Figure 2, panel A (table): Mutation data - I cannot find a corresponding section in the Methods about how these data were obtained, presumably by some form of bulk sequencing? Also, there is no analysis of how the mutations in these samples correlate to mutations in the corresponding diagnostic material - do both methods retain mutations found in the diagnostic tumour tissue material?

Page 23, Figure 2, panel B: scale bars are needed

Page 23, Figure 2, panel D: I cannot see how the same scale bar can represent 100 um in each main photomicrograph and 50 um in the insets, where the magnification seems to be > 3 times that in the corresponding main photomicrographs. Please correct

Table 1, Basal Cell Carcinoma SB 02: Sex is 'W', please define what 'W' sex is

Reviewer #3 (Remarks to the Author):

Review of the manuscript, "Live slow-frozen human tumor tissues viable for 2D, 3D, ex vivo cultures and scRNAseq"

The authors present a workflow for cryo-storing viable human tumor cells by slow-freezing. They compare the possibilities of using these viably frozen samples to freshly isolated cells to generate in vitro primary cell models. Moreover, they assess whether viably frozen material shows comparable RNA profiles to cells dissociated from matched fresh tumors.

This is an interesting presentation of how surplus tumor material can be used for cancer research. It is a valuable resource for the growing number of institutions seeking to maximize the benefits of their biobanks built around diagnostic sample acquisition or based on excess patient material. However, the

report by Restivo et al. appears to be a conglomerate of methods and the results are in many parts rather anecdotal. Moreover, the authors fail to clearly state their unique angle on the topic of cell biobanking. Rather than being an original report, this manuscript illustrates the perspective and experience of the authors' institute and the analysis confirms results from other biobanking studies.

Major points:

The sample numbers analyzed here were small:

- 2D ex vivo culture from 35 fresh and 36 low-frozen melanoma samples. It is not clear whether these were matched (fresh/frozen from the same patient).
- 3D organoid models from 2 CRCs (matched fresh/frozen).
- Shave biopsies from 2 BCC patients (matched fresh/frozen).
- scRNAseq from matched fresh and frozen specimens from 3 BCC; 2 CRC and one melanoma.
- scRNAseq from 5 frozen breast cancer samples are presented in a separate paragraph

With the exception of the melanoma samples, it is very difficult to generalize based on such small numbers of samples. Moreover, it is uncertain whether an adaptation to additional or even different tissue and tumor samples may be possible. In that context, it is also surprising that for the retrospective analysis of primary cell cultures from melanoma tissue, the authors repeatedly mention the success rate of their approaches but never present the actual numbers in the main text.

It is not clear what should be original or unique about this report. As the authors correctly point out, several other studies have previously reported very similar if not identical approaches for viable cryopreservation of patient cells or tissue combined with either patient-derived cell model generation or scRNAseq (references 1, 2, 3, 8, 9, 10, 19 and 20).

The ex vivo cell models presented in this manuscript were apparently not used for any downstream assays. Particularly for the shave biopsy, it is difficult to understand whether they can be considered a cell or tissue culture as suggested by the heading of the corresponding section or what they should be used for.

In the context of the methods performed in this study, it would make sense to analyze the ex vivo models by scRNAseq and assess which cellular populations are retained in cell cultures generated from fresh or frozen starting material. This is particularly important for the cell cultures derived from melanoma for which the authors claim to be able to evaluate fibroblast contamination by visual inspection. A detailed transcriptional profile could have confirmed this assumption and made a more solid statement about the problem of fibroblast overgrowth.

This as well as previous studies find that epithelial cells are partially lost due to the freezing process. There is no further discussion or analysis of this aspect even though it seems problematic considering that the tumor cells are of epithelial origin. Thus, frozen samples could in fact not be as representative for the original tumor cell populations. Selective pressures during cell culturing may even augment this problem. Moreover, certain types of immune cells cannot be recovered from frozen samples and it needs to be discussed how this could hinder the application of such specimens. The manuscript would benefit from a more adequate and detailed discussion of these aspects.

Minor points

Which of the authors contributed equally?

Consistent and correct terminology should be used throughout the manuscript. It is for example not common to refer to tumor tissue that has been cut into small pieces as "microtumors". "Mixability" is not an English word. Vague and imprecise descriptions such as "...certain assays.", "...different types of methods..." should be avoided.

Page 3 line 63: FFPE or snap freezing are non-viable methods for tissue preservation. It is not correct to state that the viability of cells is decreased.

What is the difference between fresh-frozen and snap frozen as stated in the abstract?

The paragraph about the quantification of the distribution of cell types by Kolmogorov-Smirnov test is very hard to understand. It is unclear why the D-statistic was employed and what this analysis signifies.

We thank the reviewers for their thoughtful comments and constructive remarks. We appreciate the effort to improve our work and have addressed every issue to improve and clarify the manuscript. To address the issues raised by the reviewers and to further support the claims made in the manuscript we analyzed public data available from a published study comparing fresh vs. slow frozen with our dataset, described in more details 2D culture as well as included further examples of ex vivo application by citing other papers from our and external groups. Please find below a detailed point-by-point our response (in red) to reviewers' comments (in black).

Reviewers' comments:

Reviewer #1 (Remarks to the Author):

Remarks to be sent to the author:

Overall comment:

The manuscript 'Live slow-frozen human tumor tissue viable for 2D, 3D, ex vivo cultures and scRNAseq' by Restivo et al., compares using fresh vs. slow-frozen tissues from various malignancies for establishing different methods such as 2D, 3D, organoids, ex vivo cultures as well as scRNAseq. The manuscript is well written and follows a logical order. It covers 3 different cancer types and explores different growth conditions including state-of-the-art organoids and scRNAseq. However, despite the authors examining the important aspect of freezing and reusing human tissues for various research applications, the manuscript lacks exploration of some important technical aspects such as the effects of tissue freezing times and the various digestion protocols used.

Reply: Thank you very much for pointing out the importance of cryopreservation duration and the effect of different digestion protocols. Within this study, while establishing scRNAseq and other applications (2D, 3D and ex vivo) on primary clinical tissue, the time passed between processing fresh and slow-frozen samples varied between 1 week to 14 months (see supplementary table 1). We have not tested cryopreservation for longer than 14 months and added a sentence into the discussion part as a limitation of our study (page 11):

'One limitation of our study is that we have not tested if slow frozen samples stored beyond 14 months have any further effect on the tissue quality and affect downstream applications. However, we can hypothesize that slow frozen tumor pieces are well preserved similar to cell lines stored for years in liquid nitrogen, provided that tissue pieces are small enough to allow DMSO penetration and stable storage temperature are guaranteed. Based on our results we can conclude that tumor pieces can be stored for up to one year as slow frozen samples for future downstream analysis..'

Regarding the different digestion protocols, they have been optimized within other projects (manuscript in revision) and for the current study, we already used optimized dissociation protocols to specifically compare the effect of cryopreservation of the tissue. In this study, we used two established protocols, for skin and tumor digestion that yield a sufficient number of cells, preserve the viability and have minor effects on transcriptome signatures. The tumor digestion protocol was established within a Tumor Profiler study that profiled metastatic melanoma, metastatic epithelial ovarian cancer, and acute myeloid leukemia metastatic tumors (Irmisch et al., 2021). Within this study, we used this protocol to dissociate CRC and liver metastasis tumors for scRNAseq and 3D organoids. The skin dissociation was optimized within a different project on slow frozen skin tissue (now under revision, confidential). We attach a table with important readouts of skin dissociation protocol (Table 1) and scRNAseq results showing the effect on the transcriptomic profile (shown by the density of the cell clusters on UMAPs from 3 tested conditions, Figure 1). From this optimization we concluded that the best viability, optimal cell number as well as preservation of transcriptomic signatures were obtained using the D/C/P protocol that was used in this study for BCC samples. Therefore, the optimized skin dissociation protocol was used in this study for BCC digestion.

Skin location	Dissociation protocol	Weight (mg)	Total isolated cell number	Viability (%)	Live cells/mg of
---------------	-----------------------	-------------	----------------------------	---------------	------------------

					tissue
Arm flap	MACS ^M	91	155 000	31	528
	Dispase I/Collagenase IV/Trypsin (D/C/P)	81	121 000	72	1076
	Dispase I/Cold Proteases (D/Cp)	54	46 000	86	733
Breast area	MACS	183	452 000	51	1260
	Dispase I/Collagenase IV/Trypsin (D/C/P)	152	457 000	67	2014
	Liberases ^{DH}	139	71 000	65	332

Table 1: Optimization of skin dissociation protocol.

Figure 1

Additional specific comments:

1. The paper does not come across as being high on innovation. The authors themselves summarize previous work that has shown similar results to what they show. It is true that the authors go beyond and show similarities on the single-cell level while focusing on the immune subtype, however, their conclusions are not always biologically clear. For example, in page 7, line 203, the authors fail to explain what the results indicate: 'FOS and heat-shock proteins in over 10 out of the 17 cell types'. It is simply not clear what this conclusion means. The authors should highlight and emphasize more the novel and innovative aspects of their work, as this is currently missing.

Reply: We thank you for your feedback. We agree that the techniques we described are not novel, but we could demonstrate their broader application on a diverse set of slow-frozen tumors. This biobanking approach, if applied routinely, can be transformative for cancer research. Currently, primarily formalin-fixed and paraffin-embedded as well as snap-frozen techniques are often utilized which cannot be applied in live-cell assays as demonstrated by our work on slow-frozen material. Additionally, within this study, we demonstrated that slow frozen *ex vivo* tissue sections retain the tumor architecture and proliferation capacity to at least a similar degree as the fresh pairs and can be thus used for downstream applications such as drug screening, which is typically performed from freshly obtained tissue biopsies as demonstrated by our group previously (Saltari et al., 2021). We added the text accordingly in the results paragraph: *Ex vivo* culture of BCC from fresh and slow frozen material (shave biopsies). “*Ex vivo* cultures of different types of cancer including melanoma, BCC and breast cancer have been successfully used for many applications including drug testing. Here we show for the first time that it is possible to slow-freeze the tissue and use it later as *ex vivo* material without affecting tumor morphology and proliferation. We thus assume that slow frozen tissue can be used subsequent to storage for drug testing.”

We thank the reviewer for pointing out unclear statements and conclusions on FOS and heat shock proteins in some cell types. We added an enrichment analysis to better explain the importance of these genes (Results section, chapter: Cell-type dependent activation of heat-stress response in slow-freezing). Showing these results help us better emphasize that the difference between fresh and frozen cells (at least in those cell types where there is a detectable difference) is in the activation of the cellular response to heat stress pathway.

Regarding other similar works published reporting comparison of fresh and slow frozen samples for scRNAseq application, we would like to highlight that we report side-by-side comparison for a set of tumors distinct from previously reported: BCC, CRC and melanoma. Work by Wu *et al.* reports a comparison between fresh and live-frozen pairs in two prostate, three breast cancer and one melanoma metastasis in LN. In addition to scRNAseq, we report side by side comparison of fresh and slow frozen tissue applied to more affordable and widely used cancer biology assays such as 2D, 3D and *ex vivo* cultured tumor slices, which has not been reported before.

2. As mentioned above, the authors do not discuss the effects of long-term freezing and whether it influences the quality of tissue, nor do they discuss digestion protocols. There have been papers in the past (e.g., Quintana, 2008) discussing how digestion protocols affects tumor formation and the number of tumor-initiating cells. Can the authors please comment and include this in their analyses? They mention this briefly in line 265, but this is not sufficient.

We thank the reviewer for pointing out those important issues. First, we added the supplementary table 1 with the duration of slow freezing for each sample that was used within this study. We haven't tested the effect of cryopreservation on cell type composition and sample viability for longer than 14 months and added a sentence into a discussion part as a limitation of our study (page 12). We want to emphasize and we added a clarification that the main purpose of this study was to compare paired fresh vs slow frozen samples.

Regarding the sentence in line 265:

“Epithelial cells (including keratinocytes) and CAFs were affected more in BCC cancer cells compared to CRC which could be due to shorter enzymatic digestion times or tissue specific characteristics.”

The dissociation protocols for tumor digestion used in this study were optimized within other projects. Due to tissue-specific composition, the digestion protocol used for CRC and its metastasis could not be applied to skin tumors such as BCC. Dispase used to digest BCCs is required to disturb collagen and fibrinogen connections (Reichard and Asosingh, 2019). Moreover, skin digestion duration is on average 2.5-3 hours, compared to tumor digestion protocol which is only up to one hour. Therefore, we cannot test if any transcriptional changes might have been introduced by different digestion protocols/enzymes used between BCC and CRC samples, and only hypothesize that duration might have caused additional changes in BCC. We acknowledge and discuss this limitation in the discussion part

We are aware of the work of Quintana *et al.* of 2010 (Quintana *et al.*, 2010) in which it was shown that using trypsin in the digestion solution important cell populations in melanoma (i.e. the CD271 positive cells) were lost and then Civenni *et al.* (Civenni *et al.*, 2011) could show that removing trypsin from the digestion protocol increased the number of those cells. Although those are very important findings, we feel they are not critical in the methods we used. For 3D cultured organoids where digestion could affect the surface epitope capture, we could demonstrate a comparable cell composition in fresh and slow-frozen starting material, based on specific CRC marker expression. Using pairs of fresh and slow frozen samples, we profiled the transcriptome using scRNAseq and stained FFPE of *ex vivo* tissue sections that required no digestion for downstream analysis. As we apply a similar digestion protocol to paired fresh and slow frozen samples and we recover similar cell types, sometimes in different proportions, we can conclude that cell type composition variations come from preservation method, and not from the digestion protocol. This information is present in the discussion paragraph : "In this study, we observed that epithelial cells (including keratinocytes) and CAFs were affected more in BCC cancer cells compared to CRC which could be due to shorter enzymatic digestion times or tissue-specific characteristics. This was even more evident in the melanoma FNA, where no digestion procedure was applied and the fewest differences between the fresh and the slow-frozen samples were detected."

Minor comments:

1. There are some typos where the authors sometimes use μ to indicate micro, and sometimes they use u
2. In p. 16, the table is missing a cat.no. It is simply written as xx.

Thank you for pointing out those inaccuracies, we fixed them in the main text.

Reviewer #2 (Remarks to the Author):

Restivo *et al.* describe a comparison of cryopreserved vs. freshly processed human tumour tissues for culture and scRNAseq, and report encouraging results indicating that cryo-preserved human tumour tissues (at least from colon cancer and basal cell carcinoma) remain representative of the original tumour material to a similar extent as freshly processed tumour tissue samples, as determined by establishing tumour tissue cultures and by performing scRNA sequencing.

After reporting comparable success rates for establishing tissue cultures from cryopreserved melanoma samples as for freshly processed ones, the authors then analyse scRNAseq data or tissue culture establishment in matched fresh vs. slow-frozen samples from each of 5 BCC (3 scRNAseq, 2 culture), 4 colon cancers (2 scRNAseq, 2 culture) and one melanoma fine needle aspirate (scRNAseq). I think that this section is the most novel part of the manuscript, and I find it particularly impressive that the authors analysed matched slow-frozen vs. fresh processed samples from each of the tumours.

In addition, the authors also studied five slow-frozen breast cancer cases by scRNAseq (i.e. without matching fresh processed material).

I was impressed by the thorough analysis of the scRNAseq in matched fresh vs. slow-frozen samples, showing similar cell-type heterogeneity, albeit with preferential loss of granulocytic cells in slow-frozen samples.

I think it would be a good idea for the authors to include a comparison of their work with that reported by He *et al.* in 2020 (Biopreservation and Biobanking, Vol 18, No. 3 p. 222-227) DOI: 10.1089/bio.2019.0062. I think Restivo *et al.*'s work compares favorably with that of He *et al.*, and it would be sensible to compare the approaches and findings in the two papers.

Reply: We thank you for your positive feedback and appreciate your valuable input. Indeed our results and the results mentioned in He *et al.* 2020, are similar in particular in showing reproducible cell viability in fresh and slow-frozen samples and success of organoids establishment from paired samples. We add this excellent work as a supportive reference in pages 6 and 10.

Some specific comments

Page 1 line 17: Who are the authors who contributed equally? I do not see any marked by the '7' in superscript

Page 3 line 65: 'showed' instead of 'have shown'

Page 3 line 65: define PDX at first use (patient-derived xenograft)

Page 4 line 94: perhaps 'broad' rather than 'broader'?

Page 4, line 110: 'compared slow-freezing methods': it seems that only one method (singular, not plural) for slow-freezing was used in this work; the comparison is with fresh processing of tumour tissue and the outcomes being tested are scRNAseq characteristics and establishment of tumour cultures

Page 13, line 402: Ventana Benchmark Ultra is made by Roche, not by Leica Bond (Leica Bond is a different platform for immunohistochemistry)

Thank you for catching those inaccuracies in the text. We corrected all the points.

Page 23, Figure 2, panel A (table): Mutation data - I cannot find a corresponding section in the Methods about how these data were obtained, presumably by some form of bulk sequencing? Also, there is no analysis of how the mutations in these samples correlate to mutations in the corresponding diagnostic material - do both methods retain mutations found in the diagnostic tumour tissue material?

Thanks for pointing out this important missing information. We added more details in the paragraphs: **“Retrospective analysis of primary cell culture establishment from fresh and slow-frozen melanoma tissue”** and in the material and methods **“Melanoma cell line isolation and culturing”**.

Briefly, the mutational state of melanoma biopsies used to isolate cells was assessed using the Melarray panel or sanger sequencing. (genes on the panel of MelArray are in Supplementary Table 3).

After cells were isolated, to confirm the mutation found in the parental tumor, DNA from cells was isolated and amplified with primers for the specific genes mutated in the parental tumor (Supplementary Table 4). After amplification, we sequenced the material using the M13 primers or primers specific for the sequence to amplify and with the 3500 genetic analyzer. The amplified products were aligned to the wild type sequences with the BLAST function of NCBI. Fibroblasts contamination was assessed by morphological examination and derived from the percentage of cells carrying the specific mutations (found in the parental tumors) in the cell culture populations (cells carrying the mutations are melanoma, cells negative for the mutations are fibroblasts). Melanoma cell cultures were considered successful when the mutations present in the parental tumors were present in the isolated cells and for at least 2 passages and if the fibroblasts contamination was less than 10%.

Page 23, Figure 2, panel B: scale bars are needed. Insert the magnification and mention in the figure legend

Thank you for catching this issue. We fixed it by inserting the magnification and mentioning it in figure legend.

Page 23, Figure 2, panel D: I cannot see how the same scale bar can represent 100 um in each main photomicrograph and 50 um in the insets, where the magnification seems to be > 3 times that in the corresponding main photomicrographs. Please correct.

Thank you for catching this issue. We fixed it by re-exporting the raw images with scale bars directly produced by the program ScanScope.

Table 1, Basal Cell Carcinoma SB 02: Sex is 'W', please define what 'W' sex is done (changed W with F

Thank you for pointing out this issue, we fixed it in the table.

Reviewer #3 (Remarks to the Author):

Review of the manuscript, "Live slow-frozen human tumor tissues viable for 2D, 3D, ex vivo cultures and scRNAseq"

The authors present a workflow for cryo-storing viable human tumor cells by slow-freezing. They compare the possibilities of using these viably frozen samples to freshly isolated cells to generate in vitro primary cell models. Moreover, they assess whether viably frozen material shows comparable RNA profiles to cells dissociated from matched fresh tumors.

This is an interesting presentation of how surplus tumor material can be used for cancer research. It is a valuable resource for the growing number of institutions seeking to maximize the benefits of their biobanks built around diagnostic sample acquisition or based on excess patient material. However, the report by Restivo et al. appears to be a conglomerate of methods and the results are in many parts rather anecdotal. Moreover, the authors fail to clearly state their unique angle on the topic of cell biobanking. Rather than being an original report, this manuscript illustrates the perspective and experience of the authors' institute and the analysis confirms results from other biobanking studies.

Major points:

The sample numbers analyzed here were small:

- 2D ex vivo culture from 35 fresh and 36 low-frozen melanoma samples. It is not clear whether these were matched (fresh/frozen from the same patient). not matched, written in the text
- 3D organoid models from 2 CRCs (matched fresh/frozen).
- Shave biopsies from 2 BCC patients (matched fresh/frozen).
- scRNAseq from matched fresh and frozen specimens from 3 BCC; 2 CRC and one melanoma.
- scRNAseq from 5 frozen breast cancer samples are presented in a separate paragraph

With the exception of the melanoma samples, it is very difficult to generalize based on such small numbers of samples. Moreover, it is uncertain whether an adaptation to additional or even different tissue and tumor samples may be possible. In that context, it is also surprising that for the retrospective analysis of primary cell cultures from melanoma tissue, the authors repeatedly mention the success rate of their approaches but never present the actual numbers in the main text

Reply: Many thanks for the important feedback. Regarding the point about the sample size, for scRNAseq other studies profiled similar or lesser number of samples. E.g. Wu *et al.* (Wu *et al.*, 2021) profiled 6 samples of other cancer types (two prostate cancers, three breast cancers and one cutaneous melanoma) therefore our cohort consisting of BCC, CRC and subcutaneous melanoma metastasis, consisting also of 6 samples, is comparable in terms of samples size and complementary in terms of type of cancers analyzed. Other studies like Denisenko *et al.* (Denisenko *et al.*, 2020) compared fresh vs. cryopreserved mouse kidney samples (n=2).

Concerning the 3D and ex vivo cultures, we would like to emphasize that for all our analysis we used surplus human tissue deriving from routine surgeries. This makes the access to this material very limited (in terms of availability and amount of usable material) because the majority of the sample has to be used for diagnostic purposes. Overall, in this study, we used 5 BCC, 4 CRC and 72 melanoma biopsies which we strongly believe makes a strong cohort of samples analyzed. The number of samples we analyzed by scRNAseq is limited as the cost of this technology prohibited the expansion of our cohort. However, we also analyzed publicly available datasets from similar experimental settings to demonstrate the reproducibility and robustness of our findings.

We shared your concern about the lack of information on primary melanoma cell lines establishment and success rate. Therefore we added a detailed description of the protocol in the paragraphs **"Retrospective analysis of primary cell culture establishment from fresh and slow-frozen**

melanoma tissue” and “Melanoma cell line isolation and culturing”. Briefly, the mutational state of melanoma biopsies used to isolate cells was assessed using the Melarray panel or sanger sequencing. (genes on the panel of MelArray are in Supplementary Table 3).

After cells were isolated, to confirm the mutation found in the parental tumor, DNA from cells was isolated and amplified with primers for the specific genes mutated in the parental tumor (Supplementary Table 4). After amplification, we sequenced the material using the M13 primers or primers specific for the sequence to amplify and with the 3500 genetic analyzer. The amplified products were aligned to the wild type sequences with the BLAST function of NCBI. Fibroblasts contamination was assessed by morphological examination and derived from the percentage of cells carrying the specific mutations (found in the parental tumors) in the cell culture populations (cells carrying the mutations are melanoma, cells negative for the mutations are fibroblasts). Melanoma cell cultures were considered successful when the mutations present in the parental tumors were present in the isolated cells and for at least 2 passages and if the fibroblasts contamination was less than 10%.

We performed a retrospective analysis on cells isolated either from fresh or slow-frozen melanoma tissue **(not-paired)**

It is not clear what should be original or unique about this report. As the authors correctly point out, several other studies have previously reported very similar if not identical approaches for viable cryopreservation of patient cells or tissue combined with either patient-derived cell model generation or scRNAseq (references 1, 2, 3, 8, 9, 10, 19 and 20). The ex vivo cell models presented in this manuscript were apparently not used for any downstream assays. Particularly for the shave biopsy, it is difficult to understand whether they can be considered a cell or tissue culture as suggested by the heading of the corresponding section or what they should be used for.

Reply: We thank you for your feedback. We acknowledge that we might have confused the reviewers about the aim of our paper. The novelty of this work is demonstrated by a wide range of applications that are routinely used by scientists in many labs, however we report that those assays can be performed from both fresh and slow frozen (viable cryopreservation), and highlight comparable results. *Ex vivo* cultured tumor sections represent an important tool for drug screening (Saltari et al., 2021), but typically performed from freshly obtained samples. Within this study, we demonstrated that slow frozen *ex vivo* tissue sections retain viability and proliferation capacity to at least similar degree as the fresh pairs and therefore could be used for downstream applications such as drug screens.

In the context of the methods performed in this study, it would make sense to analyze the ex vivo models by scRNAseq and assess which cellular populations are retained in cell cultures generated from fresh or frozen starting material.

Reply: Thank you for the suggestion. We agree that the culturing of shave biopsies from fresh or slow-frozen starting material might have an influence on cells within the tissue. However, we do not think that only 5 days in culture can drastically affect the cell type composition in the tissue, especially because the tissue was not manipulated after thawing. Moreover, we know from the scRNAseq analysis on fresh and slow frozen BCC biopsies that the tumor population is still present and with comparable characteristics in the pairs, even after digestion (see paragraph: “Single-cell RNA sequencing uncovers similar degrees of cell-type heterogeneity in fresh and slow-frozen samples”).

This is particularly important for the cell cultures derived from melanoma for which the authors claim to be able to evaluate fibroblast contamination by visual inspection. A detailed transcriptional profile could have confirmed this assumption and made a more solid statement about the problem of fibroblast overgrowth.

Reply: We absolutely agree that we overlooked this important piece of information in the initial submission and we thank you for pointing it out. As mentioned before a more detailed description of the cell culture derived from melanoma was added in the text. Paragraphs **“Retrospective analysis of**

primary cell culture establishment from fresh and slow-frozen melanoma tissue” and “Melanoma cell line isolation and culturing”

This as well as previous studies find that epithelial cells are partially lost due to the freezing process. There is no further discussion or analysis of this aspect even though it seems problematic considering that the tumor cells are of epithelial origin. Thus, frozen samples could in fact not be as representative for the original tumor cell populations. Selective pressures during cell culturing may even augment this problem. Moreover, certain types of immune cells cannot be recovered from frozen samples and it needs to be discussed how this could hinder the application of such specimens. The manuscript would benefit from a more adequate and detailed discussions of these aspects.

Reply: We agree about the fact that the slow freezing procedure could have an effect especially on epithelial cells, but even though fewer epithelial cells were recovered in slow frozen samples (compared to fresh), this population was still present and with similar comparable characteristics in both fresh and slow frozen pairs. Those results suggest that tumor tissue that underwent the freezing procedure can still be used for other downstream applications such as scRNAseq, 2D or 3D cultures when fresh processing is not possible, as demonstrated by our work.

Regarding the immune cells being more sensitive to slow freezing procedure, we have described this point in results section “Single-cell RNA sequencing uncovers similar degrees of cell-type heterogeneity in fresh and slow-frozen samples” stating that granulocytes (neutrophils and mast cells) were hardly detectable in slow frozen samples. To clarify this further we added a sentence in the same paragraph where we recommend to work with fresh material when granulocytes are the topic of the study:

“Among immune cell populations, granulocytes such as mast cells and neutrophils were hardly detectable in the slow-frozen tissues, which underlines the importance of using fresh material when studying these cell populations.”

Minor points

Which of the authors contributed equally?

Thank you, we have fixed this in the text

Consistent and correct terminology should be used throughout the manuscript. It is for example not common to refer to tumor tissue that has been cut into small pieces as “microtumors”. “Mixability” is not an English word. Vague and imprecise descriptions such as “...certain assays.”, “...different types of methods...” should be avoided.

Page 3 line 63: FFPE or snap freezing are non-viable methods for tissue preservation. It is not correct to state that the viability of cells is decreased.

Thank you for pointing out this inaccuracy, we fixed this in the text.

What is the difference between fresh-frozen and snap frozen as stated in the abstract?

With fresh- frozen we meant the not fixed tissue included directly in Tissue-Tek OCT. However, we understand and agree that this description can be confusing and we clarified this in the text by providing further details.

The paragraph about the quantification of the distribution of cell types by Kolmogorov-Smirnov test is very hard to understand. It is unclear why the D-statistic was employed and what this analysis signifies.

Thank you for your comment. We have decided to instead show the correlation coefficients between the gene expression in cell types in freshly processed and slow-frozen tissues. The difference between this analysis and using a Kolmogorov-Smirnov (KS) test is that while the former compares the average gene expression in cell types directly, the KS-test is based on the UMAP coordinates (calculated based on the gene expression) of individual cells. In order to put the correlation coefficients between fresh and frozen cells into perspective, we are also depicting the correlation between samples of different patients

suffering from the same disease. This reduces the effect cell types with more variable gene expression (such as T-cells in our dataset) would have on the analysis (which is accounted for in a KS-test).

References

- Civenni G, Walter A, Kobert N, Mihic-Probst D, Zipser M, Belloni B, et al. Human CD271-positive melanoma stem cells associated with metastasis establish tumor heterogeneity and long-term growth. *Cancer Res* 2011;71(8):3098-109.
- Denisenko E, Guo BB, Jones M, Hou R, de Kock L, Lassmann T, et al. Systematic assessment of tissue dissociation and storage biases in single-cell and single-nucleus RNA-seq workflows. *Genome Biol* 2020;21(1):130.
- Irmisch A, Bonilla X, Chevrier S, Lehmann KV, Singer F, Toussaint NC, et al. The Tumor Profiler Study: integrated, multi-omic, functional tumor profiling for clinical decision support. *Cancer Cell* 2021;39(3):288-93.
- Quintana E, Shackleton M, Foster HR, Fullen DR, Sabel MS, Johnson TM, et al. Phenotypic heterogeneity among tumorigenic melanoma cells from patients that is reversible and not hierarchically organized. *Cancer Cell* 2010;18(5):510-23.
- Reichard A, Asosingh K. Best Practices for Preparing a Single Cell Suspension from Solid Tissues for Flow Cytometry. *Cytometry A* 2019;95(2):219-26.
- Saltari A, Dzung A, Quadri M, Tiso N, Facchinello N, Hernandez-Barranco A, et al. Specific Activation of the CD271 Intracellular Domain in Combination with Chemotherapy or Targeted Therapy Inhibits Melanoma Progression. *Cancer Res* 2021;81(23):6044-57.
- Wu SZ, Roden DL, Al-Eryani G, Bartonicek N, Harvey K, Cazet AS, et al. Cryopreservation of human cancers conserves tumour heterogeneity for single-cell multi-omics analysis. *Genome Med* 2021;13(1):81.

REVIEWERS' COMMENTS:

Reviewer #1 (Remarks to the Author):

The authors addressed the comments to a satisfactory level. I have no further concerns.

Reviewer #3 (Remarks to the Author):

The authors have addressed the majority of the reviewer comments and the manuscript has improved significantly.

The authors present genetic data to prove that melanoma primary cell cultures could successfully be established. While this is a very reliable method to confirm that cultured cells retain the cancer-specific mutations, it is too far fetched to assume that all other cells must be fibroblasts. Especially since no further single cell RNAseq data is presented to clarify this point. It seems possible that also normal skin cells could for example grow during the cell culture derivation. Unless the authors can show some evidence for the claim that these "contaminating" cells are fibroblasts, they should tone down these statements.

Unfortunately, the figures lacked numbering in the pdf version of the manuscript. I could only guess which were the correct figures to accompany the text and therefore I suggest that the correctness of captions and figure references in the main text is confirmed.

We thank the reviewer #3 for the thoughtful comments and constructive remarks. Please find below our response (in red) to reviewer comments (in black).

Reviewers' comments:

Reviewer #3 (Remarks to the Author):

The authors have addressed the majority of the reviewer comments and the manuscript has improved significantly.

The authors present genetic data to prove that melanoma primary cell cultures could successfully be established. While this is a very reliable method to confirm that cultured cells retain the cancer-specific mutations, it is too far fetched to assume that all other cells must be fibroblasts. Especially since no further single cell RNAseq data is presented to clarify this point. It seems possible that also normal skin cells could for example grow during the cell culture derivation. Unless the authors can show some evidence for the claim that these "contaminating" cells are fibroblasts, they should tone down these statements.

Unfortunately, the figures lacked numbering in the pdf version of the manuscript. I could only guess which were the correct figures to accompany the text and therefore I suggest that the correctness of captions and figure references in the main text is confirmed.

We thank you for the comments. We agree that we can't be sure without a more detailed analysis that the contaminating cells are fibroblasts. We changed fibroblasts with "non-melanoma cells" throughout the main text (marked in yellow).

We now numbered the figures in the right order.